# Prevalence of depression in Uganda: A systematic review and meta-analysis

**Mark Mohan Kaggwa**[1,2,3]\*, **Sarah Maria Najjuka**[4], **Felix Bongomin**[5], **Mohammed A. Mamun**[6,7], **Mark D. Griffiths**[8]

**1** Department of Psychiatry, Mbarara University of Science & Technology, Mbarara, Uganda, **2** African Centre for Suicide Prevention and Research, Mbarara, Uganda, **3** Department of Psychiatry and Behavioural Neurosciences, Forensic Psychiatry Program, McMaster University, Hamilton, Ontario, Canada, **4** School of Medicine, College of Health Sciences, Makerere University, Kampala, Uganda, **5** Department of Medical Microbiology & Immunology, Faculty of Medicine, Gulu University, Gulu, Uganda, **6** CHINTA Research Bangladesh, Savar, Dhaka, Bangladesh, **7** Department of Public Health and Informatics, Jahangirnagar University, Savar, Dhaka, Bangladesh, **8** Psychology Department, Nottingham Trent University, Nottingham, United Kingdom

\* kmarkmohan@gmail.com

## Abstract

### Background

Depression is one of the most studied mental health disorders, with varying prevalence rates reported across study populations in Uganda. A systematic review and meta-analysis was carried out to determine the pooled prevalence of depression and the prevalence of depression across different study populations in the country.

### Methods

Papers for the review were retrieved from *PubMed*, *Scopus*, *PsycINFO*, *African Journal OnLine*, and *Google Scholar* databases. All included papers were observational studies regarding depression prevalence in Uganda, published before September 2021. The Joanna Briggs Institute Checklist for Prevalence Studies was used to evaluate the risk of bias and quality of the included papers, and depression pooled prevalence was determined using a random-effects meta-analysis.

### Results

A total of 127 studies comprising 123,859 individuals were identified. Most studies were conducted among individuals living with HIV ($n = 43$; 33.9%), and the most frequently used instrument for assessing depression was the Depression sub-section of the Hopkins Symptom Checklist ($n = 34$). The pooled prevalence of depression was 30.2% (95% confidence interval [CI]: 26.7–34.1, $I^2 = 99.80$, $p<0.001$). The prevalence of depression was higher during the COVID-19 pandemic than during the pre-pandemic period (48.1% vs. 29.3%, $p = 0.021$). Refugees had the highest prevalence of depression (67.6%; eight studies), followed by war victims (36.0%; 12 studies), individuals living with HIV (28.2%; 43 studies), postpartum or pregnant mothers (26.9%; seven studies), university students (26.9%; four studies),

**Data Availability Statement:** https://doi.org/10.6084/m9.figshare.19579096.v1.

**Funding:** The author(s) received no specific funding for this work.

 

**Competing interests:** The authors have declared that no competing interests exist.

children and adolescents (23.6%; 10 studies), and caregivers of patients (18.5%; six studies).

## Limitation

Significantly high levels of heterogeneity among the studies included.

## Conclusion

Almost one in three individuals in Uganda has depression, with the refugee population being disproportionately affected. Targeted models for depression screening and management across various populations across the country are recommended.

## Trial registration

Protocol registered with PROSPERO (CRD42022310122).

## Introduction

The global prevalence rates of depression and other mental disorders have been increasing since 1990 [1], and approximately 3.8% of individuals worldwide have depression [2]. The prevalence of depression increased by an additional 27.6% globally during the coronavirus disease 2019 (COVID-19) pandemic (2020 to 2021) and by 23% in sub-Saharan Africa [3]. Despite the lower increase in the burden of depression in sub-Saharan Africa, most individuals with depression go undiagnosed and untreated in the region [4]. This has led to some individuals having extreme complications of depression, such as suicide [5–11]. The effect is even worse in low-income countries such as Uganda, with a high prevalence of depression (i.e., approximately one-fifth of the population between 2010 and 2017, and 27% of outpatients being depressed based on previous systematic review and meta-analyses [12, 13]).

In Uganda, due to the high burden of depression, various studies have been conducted among different populations (e.g., infected with human immunodeficiency virus [HIV], women, cancer patients, caregivers of patients, students, etc.) to understand its effects and design possible interventions [14–17]. Ugandan clinicians and researchers have employed various psychometrically validated tools to screen and diagnose depression among Ugandans, including the Patient Health Questionnaire (PHQ), Beck's Depression Inventory (BDI), Hamilton Rating Scale for Depression, Symptom Checklist-20, Center for Epidemiologic Studies—Depression Scale, Akena Visual Depression Inventory, and the Mini-International Neuropsychiatric Interview (MINI) [18–24].

Uganda is a landlocked low-income country that has been affected by multiple adverse events, including civil wars, extreme poverty, high rates of HIV, various epidemics (e.g., Ebola), and poor mental health services, as well as being one of the largest refugee-hosting countries in the world [25–30]. These adverse events put many Ugandans at risk of developing depression due to the multiple physical, psychological, emotional, and social difficulties they are associated with. The effects of these difficulties are evidenced by the high levels of depression within various study groups in the country, such as women, children, students, individuals living with HIV, refugees, and members of the general public, with many groups reporting depression prevalence rates of over 70% [14, 31–37].

Due to the high burden of depression and a large amount of literature concerning mental health in Uganda, various systematic reviews have been conducted, especially among individuals living with HIV [12, 15, 38]. However, a comprehensive systematic synthesis of all published literature on depression in Uganda is lacking. Therefore, the present systematic review and meta-analysis aimed to determine the pooled prevalence of depression in Uganda and determine the prevalence of depression among various study populations in the country.

## Methods

This review was conducted in accordance with the Preferred Reporting Items for Systematic Reviews and Meta-Analyses (PRISMA) guidelines [39] and the Meta-analysis of Observational Studies in Epidemiology (MOOSE) guidelines for systematic reviews and meta-analysis of observational studies [40]. The study protocol was prospectively registered with PROSPERO (CRD42022310122). The review question was formed according to the Joanna Briggs Institute (JBI) Checklist for Prevalence Studies and the CoCoPop (Condition, Context, and Population) [41]. The *condition* was depression, the *context* was Uganda, and the *population* was all studied groups. Therefore, the review's research questions were: (i) "What is the prevalence of depression in Uganda?" and (ii) "What is the prevalence of depression among various study populations in Uganda?"

### Search strategy

With the help of the university Librarian at Mbarara University of Science and Technology, relevant databases were used for the literature search, including *PubMed*, *Scopus*, *PsychInfo*, and *Google Scholar* (in the present review, eligible papers from the first 100 pages based on relevance were included), and *African Journal OnLine* (*AJOL*). The study included articles (both peer-reviewed and preprints) in all languages from 1972 (the first published paper about depression in Uganda [42]) to September 2021. The following key words were used in the literature search from different databases: (i) 'depression,' (ii) 'Uganda,' and (iv) 'prevalence' or 'incidence.' All systematic reviews concerning depression in Uganda, East Africa, and Africa were reviewed for eligible studies. The PRISMA 2020 flow chart shows the details of the search hits retrieved, included, or excluded papers [43] (Fig 1).

### Inclusion criteria and exclusion criteria

The literature search included all observational studies (cross-sectional, case-control, and cohort studies) published in all languages regarding depression prevalence (based on different assessment tools or subjective reports) in Uganda, based on various assessment tools and cut-offs; and excluded case reports, case series, qualitative research, letters to the editor, commentaries, conference proceedings or abstracts, policy papers, protocols, reviews, and meta-analyses.

### Study and data management

All identified papers were entered into *EndNote 9* to ascertain duplicates. After removing duplicates and review articles, the titles and abstracts of the articles were screened for inclusion independently by two team members. MAM settled any discrepancies after a discussion with the two team members for the reason of exclusion. The final selected papers were read for the full review, and this was done in pairs after dividing the results into two. Two members of the research team reviewed the first half, and another two members of the research team reviewed the second half. For papers whose full texts were not fully accessible, the corresponding

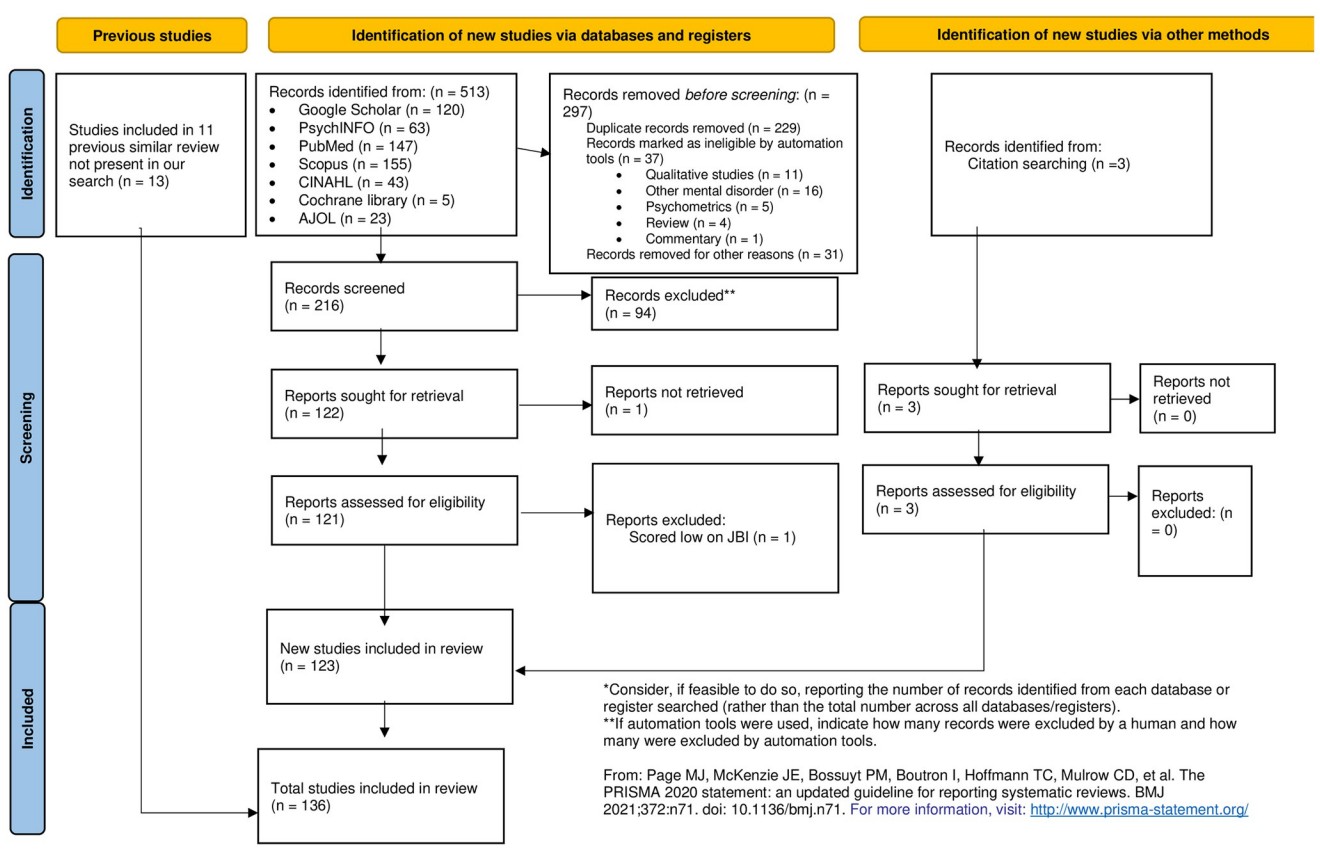

**Fig 1. The PRISMA flow chart.**

authors were contacted by email. The papers were assessed for a quality check using the JBI Checklist for Prevalence Studies [44], as used in other systematic reviews [45]. Finally, a research team member screened for eligible papers from the systematic reviews regarding depression in the region.

## Data extraction

A pre-piloted and self-designed *Google Forms* document with the following information was used to collate the data. The data extracted included: the first author, year of data collection, study design, paper quality assessment questions based on the JBI Checklist, study group, sample size, age of participants, number of male and female participants, tools/questions used to assess for depression, and prevalence of depression.

## Bias evaluation and quality assessment of the included papers

The nine-item JBI Checklist was used to evaluate the risk of bias and the quality of the included papers [44]. The JBI Checklist uses a four-point response system: "no," "yes," "unclear," and "not applicable" for the following study characteristics: (i) appropriateness of the sample frame; (ii) recruitment procedure; (iii) adequacy of the sample size; (iv) description of participants and setting; (v) description of the identified sample; (vi) validity of the methods used to screen for depression; (vii) reliability of the methods used to screen for depression; (viii) adequacy of statistical analyses; and (ix) response rate. Articles were assigned one point for each

'*yes*' response, and the remaining responses were assigned zero points. Therefore, the total score ranged from 0 to 9. Studies with a score of 4 or above were considered good quality. One article was excluded due to scoring poorly on most parameters [42]. The scores of the papers are presented in Table 1.

### Ethical considerations

The present study was a secondary analysis of previously published literature. Therefore, formal clearance by a Research and Ethics Committee was not required.

### Data synthesis and analysis

Data for this analysis are available at figshare [46]. *Microsoft Office 2016* (Microsoft Inc., Washington, USA) and *STATA 16.0 software* (Stata Corp LLC, College Station, Texas, USA) were used for data cleaning and statistical analysis. Descriptive statistics and qualitative narrative analysis were used to present individual study and participant characteristics. A random-effects meta-analysis [47] was performed using the *meta* command to determine the pooled prevalence of depression and prevalence of depression in the different study groups. The results were presented on forest plots. The Higgins Inconsistency index ($I^2$) and univariate random effect meta-regression [48] were used to evaluate the heterogeneity among the selected studies. Publication bias was assessed visually using funnel plots symmetry [49], and fill and trim estimated the number of missing studies [50]. Egger's test was also used to assess for small study effects. Univariate meta-regression was used to determine the source of heterogeneity based on the following: mean age, number per gender (males or females), data collection period (pre-COVID-19 pandemic vs. during the pandemic), study design, JBI Checklist score, sample size, and depression assessment tool used. Subgroup analysis was also conducted based on study types, study tools used, the diagnostic status of the tool, and the data collection period.

## Results

A total of 136 papers met the criteria for inclusion in the review (comprising three theses [51–53], two preprints [36, 54], and 131 peer-reviewed journal papers). Using *Microsoft Excel 2016*, duplicate papers were automatically identified [including republished datasets in different papers] (n = 9) based on year of data collection, type of study, district of study, sample size, study population, the prevalence of depression, and assessment tool used for depression. The remaining 127 papers, comprising a total of 123,859 individuals, comprised the total study sample (Table 1).

The identified papers were published between 2004 and 2021, and the data included were collected between 2000 and 2021 from 45 districts in Uganda.

Most of the studies were conducted in the capital city, Kampala (n = 43), followed by the districts of Mbarara (n = 23) and Gulu (n = 16). The pooled mean age of the participants was 27.19 years (95% CI: 24.31–30.09 years; $I^2$ = 85.38, $p<0.001$). A total of eight studies were conducted during the COVID-19 pandemic [6, 14, 16, 33, 53, 168–171].

### Tools used in assessing depression

Both diagnostic and screening tools were used to assess depression among different populations in Uganda (Table 1). The tools used included: (i) Depression sub-section of the Hopkins Symptom Checklist (DHSCL) (*n* = 34, 26.8%) [35, 55, 61, 63, 65–68, 71, 73, 74, 79–82, 84, 92, 100, 108, 110, 111, 113, 125, 135, 136, 138, 140, 141, 144, 147, 151, 155, 166], (ii) Patient Health

**Table 1. Participants' characteristics of all studies included in the meta-analysis.**

| First author, year of publication | Study design (JBI Checklist score) | Year of data collection | Districts | Study group | Sample size | Female (male) | Age (in years) | Tools used to access depression (cutoff) | Depression n (%) |
|---|---|---|---|---|---|---|---|---|---|
| Bolton 2004 [55] | CS (9) | 2000 | Masaka and Rakai | General population | 587 | 364 (223) | 39.3±2.9 | Depression section of the Hopkins Symptom Checklist (DHSCL) | 123 (21) |
| Ovuga 2006 [56] | CS (7) | 2000 and 2002 | Kampala | Non-medical undergraduate students in 2000 and undergraduate medical students in 2002 | 253 and 101 | 92 (161) and 31 (70) | 21.3± 2.4 and 23.5± 5.0 | Beck Depression Inventory (BDI)–(10) | 37 (16.2) and 4 (4) |
| Nakasujja 2007 [57] | CS (9) | 2001 | Kampala | Elderly patients in non-psychiatric wards | 127 | 64 (63) | | Self-Reporting Questionnaire (SRQ 25)–(>5) | 23 (18) |
| Nalugya-Sserunjogi 2016 [58] | CS (9) | 2003 | Mukono | school-going adolescents | 519 | 218 (301) | 16±2.18 | Children Depression Inventory (CDI)–(19) MINI-KID | 109 (21) and 8 (2.9) |
| Nakku 2006 [59] | CS (9) | 2002–2003 | Kampala | Women 6 weeks postpartum | 523 | 523 | 23.4±4.76 | MINI and SRQ-25–(>5) | 32 (6.1) and 38 (7.3) |
| Kaharuza 2006 [60] | RCT (9) | 2003–2004 | Bugiri, Busia, Mbale, and Tororo | Individuals living with HIV | 1017 | 781 (236) | 31–40 = 452; 31–40 = 452; 41–50 = 291; and 50 + = 94 | Center for Epidemiological Studies Depression scale (CES-D) | 476 (47) |
| Kinyanda 2011b [61] | CS (9) | 2003–2004 | Adjumani, Apac, Arua, Bugiri, Bushenyi, Kaberamaido, Kapchorwa, Katakwi, Lira, Moyo, Mubende, Nebbi, Soroti, and Yumbe | General populations | 4,660 | | | DHSCL–(>1.75) | 1366 (29.3) |
| Ovuga 2005 [62] | CS (9) | | Adjumani, Bugiri | General populations | 524 | | | BDI | 91 (17.4) |
| Lundberg 2011 [63] | CS (9) | 2004–2005 | Kampala, Mbarara | Residents from urban and semi-urban Kampala and Mbarara | 630 | 312 (334) | 18–24 = 337; 25–30 = 437 | DHSCL–(>1.75) | 96 (9.84) |
| Nakimuli-Mpungu 2013a [64] | CS (8) | 2004–2005 | Kiruhura | Individuals living with HIV | 244 | 156 (88) | 36.2±8.9; Range = 18–60 | DSMIV | 97 (40) |
| Agardh 2012 [65] | CS (8) | 2005 | Mbarara | Undergraduate university students at MUST | 976 | 362 (614) | Median age = 23. Younger ≤23: 628. Older>23: 329 (34.4), missing 23 | DHSCL-25—(31) | 146 (15) |
| Vinck 2007 [66] | CS (9) | 2005 | Gulu, Kitgum, Lira, and Soroti | War victims in displacement camps | 2585 | 1293 (1292) | 36.7±13.8 | DHSCL–(40) | 1151 (44.5) |
| Pfeiffer 2011 [67] | CS (7) | 2005 | Gulu | War victims | 72 | 31 (41) | Median = 23.7 | DHSCL–(>1.75) | 51 (71) |
| Martinez 2008 [68] | CS (9) | 2005 | Mbarara | Individuals living with HIV | 421 | 266 (155) | Median = 36, IQR = 11 | DHSCL–(>1.75) | 79 (18.8) |

(*Continued*)

**Table 1.** (Continued)

| First author, year of publication | Study design (JBI Checklist score) | Year of data collection | Districts | Study group | Sample size | Female (male) | Age (in years) | Tools used to access depression (cutoff) | Depression n (%) |
|---|---|---|---|---|---|---|---|---|---|
| Muhwezi 2007 [69] | CS (9) | | Kampala, Mpigi, and Mubende | Individuals attending primary health care facilities | 199 | 119 (80) | | MINI | 74 (31.6) |
| Petrushkin 2005 [70] | CS (7) | | Kampala | Individuals living with HIV | 46 | 24 (22) | 36.6 | MINI | 25 (54.35) |
| Psaros 2015 [71] | Cohort (9) | 2005–2010 | Mbarara | Uganda AIDS Rural Treatment Outcomes (UARTO) cohort study | 453 | 314 (139) | 34.9±8.3 | DHSCL–(>1.75) | 172 (37.97) |
| Nakasujja 2010 [72] | CS (8) | 2005–2007 | Kampala | HIV-positive patients are at risk for cognitive impairment, and HIV-negative patients | 127 | 84 (43) | | CES-D | 62 (48.82) |
| Kaida 2014 [73] | Cohort (9) | 2005–2012 | Mbarara | Pregnant and postpartum HIV-positive women | 447 | 447 (0) | Median = 32; IQR = 10 | DHSCL–(>1.75) | 173 (38.9) |
| Roberts 2008 [74] | CS (9) | 2006 | Amuru, Gulu | War victims | 1210 | 727 (483) | 35.4 | DHSCL–(>1.75) | 811 (67) |
| Klasen 2013 [75] | CS (9) | 2006 | Gulu | Child soldiers (children and adolescents of war abducted victims who became soldiers) | 330 | 160 (170) | 14.44±1.57 | MINI-KID | 120 (36.4) |
| Klasen 2010a [76] | CS (9) | 2006 | Gulu | Child soldiers (children and adolescents of war abducted victims who became soldiers) | 330 | 160 (170) | 14.44±1.57 | MINI-KID | 120 (36.4) |
| Klasen 2010b [77] | CS (9) | 2006 | Gulu | Child soldiers (children and adolescents of war abducted victims who became soldiers) | 330 | 160 (170) | 14.44±1.57 | MINI-KID | 120 (36.4) |
| Abbo 2009 [78] | CS (9) | 2007 | Iganga and Jinja | Clients of traditional healers | 387 | 208 (178) | 34.8±13.55 | MINI | 21 (5.4) |
| Hatcher 2012 [79] | Secondary data analysis (9) | 2007 | Mbarara | HIV-infected women | 270 | 270 (0) | Median = 34. IQR = 10 | DHSCL–(>1.75) | 64 (6.7) |
| Pham 2009 [80] | CS (9) | 2007 | Amuria, Amuru, Gulu, Kitgum, Lira, Oyam, and Pader | War victims | 2875 | 1417 (1458) | 35.4±14.35 | DHSCL–(42) | 1150 (40) |
| Tsai 2012 [81] | Cohort (9) | 2007–2010 | Mbarara | Individuals living with HIV | 456 | 324 (132) | | DHSCL–(>1.75) | 71 (15.57) |
| Tsai 2016 [82] | Cohort (9) | 2007–2011 | Mbarara | Individuals living with HIV | 173 | 173 (0) | Median = 32; IQR = 11 | DHSCL–(>1.75) | 57 (33) |

(*Continued*)

**Table 1.** (Continued)

| First author, year of publication | Study design (JBI Checklist score) | Year of data collection | Districts | Study group | Sample size | Female (male) | Age (in years) | Tools used to access depression (cutoff) | Depression n (%) |
|---|---|---|---|---|---|---|---|---|---|
| Nakimuli-Mpungu 2013b [83] | A prospective study (9) | 2007–2012 | Gulu, Kitgum, Soroti, Tororo | War victim with post-trauma disorder | 2868 | | | SRQ-20 | 1297 (45.22) |
| Winkler 2015 [84] | CS (9) | 2008 | Gulu, Kitgum, and Lira | child soldiers and war-affected victims | 843 | 355 (488) | 19.0±2.7. | DHSCL–(>2.65) | 64 (7.6) |
| Musisi 2009 [85] | CS (8) | | Kampala | HIV positive adolescents | 82 | 46 (36) | 13.4±1.8 | SRQ-25 | 34 (40.8) |
| Nsereko 2018 [51]–Thesis | CS (9) | 2018 | Kampala | School adolescents | 549 | 317 (228) | | The Youth Self-Report (YSR) | 115 (21.1) |
| Wagner 2012a [86] | Cohort (9) | 2008–2009 | Jinja, Kampala | Individuals living with HIV | 602 | 409 (193) | 35.7 | Patient Health Questionnaire– 9 (PHQ-9)–(10) | 78 (13) |
| Shumba 2013 [87] | CS (7) | 2008–2009 | | Individuals living with HIV | 732 | 504 (228) | 19–39 = 294; 40–50 = 290; 50+ = 91 | Developed own tool | 432 (59) |
| Okeke 2013 [88] | Cohort (9) | 2008–2010 | Kampala | Individuals living with HIV | 482 | | 34.60±8.51 | PHQ-9 –(10) | 40 (8.3) |
| Wagner 2014a [89] | Cohort (9) | 2008–2011 | Jinja, Kampala, Mityana, and Mukono | 3 study cohorts of Individuals living with HIV | 750 | 435 (315) | 34.5 | PHQ-9 –(10) | 45 (6) |
| Wagner 2017a [90] | Cohort (9) | 2008–2011 | Kampala, Mityana, Mukono, and Wakiso | Individuals living with HIV | 1021 | 653 (368) | 36 | PHQ-9 –(10) | 92 (9) |
| Wang 2018 [91] | CS (6) | 2009 | Wakiso | Individuals living with HIV | 981 | | | Self-report to a question, "*During the last 12 months, have you had a period lasting several days when you felt sad, empty, or depressed?*" | 221 (22.5) |
| Ager 2012 [92] | CS (9) | 2009 | Amuru, Gulu | National Humanitarian Aid Workers | 376 | 134 (238) | 30.88±6.60 | DHSCL–(>1.75) | 256 (68) |
| Wagner 2011 [93] | CS (9) | | Jinja, Kampala | New HIV clients attending the clinics | 602 | 410 (192) | 36; Range = 20–62 | PHQ-9 –(10) | 78 (13) |
| Ngo 2015 [34] | Cohort (9) | 2009–2011 | Mityana, Mukono, and Wakiso | Individuals living with HIV | 1903 | 1492 (411) | 36±9 | PHQ-9–10 and MINI | 1604 (84.3) |
| Kinyanda 2011a [94] | CS (9) | 2010 | Wakiso | Individuals living with HIV | 618 | 449 (169) | 18-24yrs = 58; 25–34 = 238; 35–44 = 217; and >44 = 103 | MINI | 50 (8.1) |
| Kinyanda 2012 [95] | CS (9) | 2010 | Wakiso | Individuals living with HIV | 618 | 449 (169) | 18-24yrs = 58; 25–34 = 238; 35–44 = 217; and >44 = 103 | MINI | 50 (8.1) |
| Morof 2014 [35] | CS (9) | 2010 | Kampala | female urban refugees | 117 | 117 (0) | 31.6±4.7 | DHSCL–(>1.75) and DHSCL–(>2.65) | 112 (92) and 70 (54) |

(*Continued*)

**Table 1.** (Continued)

| First author, year of publication | Study design (JBI Checklist score) | Year of data collection | Districts | Study group | Sample size | Female (male) | Age (in years) | Tools used to access depression (cutoff) | Depression n (%) |
|---|---|---|---|---|---|---|---|---|---|
| Nakku 2013 [96] | CS (9) | 2010 | Wakiso | Individuals living with HIV | 618 | 449 (169) | 18-24yrs = 58; 25–34 = 238; 35–44 = 217; and >44 = 103 | MINI | 50 (8.1) |
| Kakyo 2012 [97] | CS (9) | | Kabarole | Postpartum mothers | 202 | 202 (0) | 24±4.33 | Edinburgh postpartum depression scale (EPDS)–(10) | 87 (43) |
| Wagner 2012b [98] | Cohort (9) | 2010–2011 | Kampala, Mityana, and Mukono | Individuals living with HIV | 798 | 530 (268) | 36.1±9.5 | MINI and PHQ-9 – (10) | 111 (13.9) and 187 (.23.43) |
| Wagner 2014b [99] | Cohort (9) | 2010–2011 | Jinja, Kampala | Individuals living with HIV | 1731 | 1131 (600) | 36 | PHQ-9 –(10) | 156 (9) |
| Kiene 2018 [100] | Cohort (9) | 2010–2011 | Butambala | HIV positive and HIV negative clients | 244 | 122 (122) | HIV positive = 34.60 ±9. 03; HIV negative = 33.28 ±10.12 | DHSCL–(>1.75) | 117 (48.0) |
| Okello 2015 [101] | Cohort (9) | 2010–2011 | Kampala | Individuals living with HIV | 798 | 530 (268) | 36.1± 9.53 | PHQ-9 –(10) | 100 (12.5) |
| Musisi 2014 [102] | Cohort (9) | 2010–2011 | Mukono, Wakiso, Kampala, and Mityana | Individuals living with HIV | 386 | 225 (161) | 35.7±8.7 | PHQ-9 –(10) | 116 (0.3) |
| Akena 2012 [103] | CS (9) | 2011 | Kampala | Individuals living with HIV | 368 | 265 (103) | 38.8±9.81; range = 18–71. | MINI | 64 (17.4) |
| Akena 2013 [104] | CS (9) | 2011 | Kampala | Individuals living with HIV | 735 | 525 (210) | 38±10.08, range = 18–71 | MINI | 72 (9.8) |
| Nakimuli-Mpungu 2011a [105] | CS (9) | 2011 | Mubende | Individuals living with HIV | 500 | 349 (151) | 40±10.7, range: = 18–80 | MINI | 230 (46) |
| Katende 2017 [106] | CS (7) | | Kampala | Caregivers of cancer patients | 119 | 79 (40) | 33±10.69 | Hospital Anxiety and Depression Scale (HADS) standardized tool | |
| Kinyanda 2013 [107] | CS (9) | | Gulu, Kaberamaido, Lira, and Tororo | Children and adolescent | 1587 | 853 (734) | ≤5 = 286; 6–9 = 416; 10–13 = 550; 14–19 = 335 | MINI-KID | 136 (8.6) |
| Spittal 2018 [108] | Cohort (9) | 2011–2012 | Amuru, Gulu, Nwoya | War victims—The Cango Lyec (Healing the Elephant) Project members 13 to 49 years who were sexually active | 2008 | 1189 (819) | Range 13–49 | DHSCL–(>1.75) | 337 (16.78) |
| Whyte 2015 [109] | CS (5) | 2011–2012 | Agago, Amuru, Gulu, and Nwoya | OPD patients in several districts in northern Uganda | 11325 | | | Clinician diagnosis | 40 (0.4) |
| Perkins 2018 [110] | CS (9) | 2011–2012 | Mbarara | General populations | 1499 | 822 (677) | below 30 = 640 | DHSCL–(>1.75) | 268 (17.88) |

(*Continued*)

**Table 1.** (Continued)

| First author, year of publication | Study design (JBI Checklist score) | Year of data collection | Districts | Study group | Sample size | Female (male) | Age (in years) | Tools used to access depression (cutoff) | Depression n (%) |
|---|---|---|---|---|---|---|---|---|---|
| Malamba 2016 [111] | Cohort (9) | 2011–2012 | Gulu, Nwoya, and Amuru | War victims—The Cango Lyec (Healing the Elephant) Project members 13 to 49 years | 2388 | 1397 (991) | Range 13–49 | DHSCL–(>1.75) | 360 (14.9) |
| Hakim 2019 [112] | CS (9) | 2011–2013 | Kampala, Wakiso | All clients aged 13 or older attending Mildmay for client-initiated HIV testing and counseling | counseling | 6998 (5234) | 13–19 = 95; 20–24 = 2607; 25–34 = 5151; 35–49 = 3010; 50+ = 510 | PHQ-2 –(4) | 1884 (15.4) |
| Familiar 2019 [113] | Cohort (9) | 2011–2014 | Soroti | HIV-positive women attending a clinic | 288 | 288 (0) | 33.5; Range = 18–54 | DHSCL–(>1.75) | 139 (61) |
| Cavazos-Rehg 2020 [114] | RCT (9) | 2012–2017 | Mbarara | HIV positive adolescents | 592 | 335 (257) | 12.13±0.65 | CDI | 298 (52.29) |
| Akena 2015 [115] | CS (9) | 2013 | Gulu, Kampala, and Mbarara | Patients with diabetes Mellitus | 437 | 283 (154) | 51±14.06, Range = 18–90 | MINI | 154 (34.8) |
| Familiar 2021 [116] | CS (9) | 2013 | Kampala | Self-settled Democratic Republic of Congo female refugees in Kampala | 580 | 580 (0) | Mean = 33.7. 18–24 = 121; 25–34 = 210; 35+ = 249 | PHQ-2 | 331 (57) |
| Akena 2016 [117] | Cohort (9) | 2013 | Luweero, Mityana, and Mpigi, Wakiso | Individuals living with HIV | 1252 | 961 (291) | 40±11. Range = 18–85 | PHQ-9 –(10) | 200 (67) |
| Rathod 2018 [118] | CS (9) | 2013 | Kamuli | General populations | 1893 | 1500 (393) | Median = 28; IQR = 15 | PHQ-9 –(10) | 80 (4.2) |
| Mugisha 2016 [28] | CS (9) | 2013 | Amuru, Gulu, and Nwoya | Individuals in post-conflict northern Uganda | 2361 | 1475 (886) | 49% of respondents were aged above 34 years | MINI | 599 (24.9) |
| Mugisha 2015 [29] | CS (9) | 2013 | Amuru, Gulu, and Nwoya | Community in the post-conflict northern Uganda | 2361 | 1475 (886) | 49% of respondents were aged above 34 years | MINI | 599 (24.7) |
| Mwesiga 2015 [119] | CS (9) | 2013 | Kampala | Individuals living with HIV | 345 | 245 (100) | Median = 35; (IQR = 12) | MINI | 17 (5) |
| Nakku 2019 [120] | CS (9) | 2013 | Kamuli | General populations | 1290 | 867 (423) | 16–30 = 494; 31–49 = 555; ≥ 50 = 240 | PHQ-9 –(10) | 325 (25.4) |
| Nalwadda 2018 [121] | CS (9) | 2013 | Kamuli | The male population in the region | 1129 | 0 (1129) | 33% of participants were below 30 years | PHQ-9 –(10) | 292 (25.9) |
| Jones 2017 [122] | Cohort (7) | 2013 | Kampala | Post-tuberculosis lung disease patients | 29 | 14 (15) | 45±13; Range = 17–69 | PHQ-9 –(5) | 7 (24) |
| Wagner 2017b [123] | Cohort (9) | 2013 | Luweero, Mityana, Mpigi, Wakiso | Individuals living with HIV | 1252 | 976 (276) | 39.8±11.2 | PHQ-9 –(10) | 375 (30) |
| Rukundo 2013 [124] | CS (9) | | Mbarara | Physically ill patients | 258 | 120 (138) | 18–24 = 50, 25–40 = 130, 41–60 = 52 | MINI | 87 (33.7) |

(*Continued*)

**Table 1.** (*Continued*)

| First author, year of publication | Study design (JBI Checklist score) | Year of data collection | Districts | Study group | Sample size | Female (male) | Age (in years) | Tools used to access depression (cutoff) | Depression n (%) |
|---|---|---|---|---|---|---|---|---|---|
| Olema 2014 [125] | CS (9) | | Gulu | Adolescents and their parents in GULU | 300 | | | DHSCL–(>1.75) | 120 (40) |
| Huang 2017 [126] | CS (9) | 2013–2014 | | Parents of children in primary school | 303 | 248 (55) | 35.92±9.80; Range = 18–79 | PHQ-9 –(10) | 85 (28) |
| Wagner 2016a [127] | Cohort (9) | 2013–2014 | Luweero, Mityana, Mpigi, and Wakiso | Individuals living with HIV | 1252 | 962 (290) | 40±11.2 | PHQ-9 –(10) | 375 (30) |
| Wagner 2016b [128] | Cohort (9) | 2013–2014 | Luweero, Mityana, Mpigi, and Wakiso | Individuals living with HIV | 1252 | 962 (290) | 40±11.2 | PHQ-9 –(10) | 375 (30) |
| Henry 2019 [129] | CS (7) | 2013–2015 | Kampala | Adolescents attending an adolescent health clinic | 514 | 372 (142) | 16; Range = 10–19 | Investigator designed tool (not standard) | 174 (33.9) |
| Meffert 2019 [15] | Cohort (9) | 2013–2017 | | Individuals living with HIV | 475 | 282 (157) | 18–33 = 146; 34–48 = 227; 49–63 = 69; 65+ = 5 | CES-D | 104 (22) |
| Swahn 2021 [130] | Cohort (9) | 2014 | Kampala | Individuals living with HIV—comparing HIV positive and non-HIV positive youth (12–18 years) | 1096 | 614 (481) | 12–14 = 219; 15–16 = 291; 17–18 = 586 | Self-report—"*In the past year, did you ever feel so sad or hopeless almost every day for two weeks or more in a row that you stopped doing your usual activities?*" | 673 (62) |
| Gyagenda 2015 [131] | CS (9) | 2014 | Kampala | Post-stroke patients | 73 | 43 (30) | 20–39 = 6; 40–59 = 25; 60–79 = 35; and 80–99 = 7 | PHQ-9 –(10) | 23 (31.5) |
| Fisher 2019 [31] | CS (6) | 2014 | Kisiro | Women attending OPD | 115 | 115 | | SRQ-20 | 87 (75.65) |
| Kinyanda 2020 [132] | CS (9) | 2014 | Kampala, Masaka | caregivers of patients with HIV | 1336 | | | Child and Adolescent Symptom Inventory-5 (CASI-5) | Baseline = 66 (5.0); 6month = 47 (4.2); 12months = 43 (4.4) |
| Muhammad 2018 [133] | CS (6) | 2014 | Kabarole | Individuals living with HIV | 150 | 84 (66) | 16.53±5.24 | - | 37 (47) |
| Nakku 2016 [21] | CS (9) | 2014 | Kamuli | Patients attending a health facility | 1415 | 1017 (398) | 33±13 | PHQ-9 –(8) | 140 (10) |
| Ashaba 2015 [134] | Case-control (9) | 2014 | Mbarara | Mother of malnourished children | 172 | 172 (0) | 25±4.43; Range = 18–40 | MINI | 46 (26.7) |
| Kinyanda 2016a [135] | CS (9) | | Amuria, Katakwi | Post-conflict communities | 1110 | 631 (479) | 56% were aged between 18 to 44 years. | DHSCL–(>1.75) | 462 (41.6) |

(*Continued*)

**Table 1.** (Continued)

| First author, year of publication | Study design (JBI Checklist score) | Year of data collection | Districts | Study group | Sample size | Female (male) | Age (in years) | Tools used to access depression (cutoff) | Depression n (%) |
|---|---|---|---|---|---|---|---|---|---|
| Manne-Goehler 2019 [136] | Cohort (9) | | Mbarara | Older individuals (above 40 years) living with HIV and HIV uninfected individuals of similar sex | 296 | 141 (155) | 52 | DHSCL–(>1.75) | 81 (27.36) |
| Kinyanda 2016b [137] | CS (9) | | | Caregivers of patients with mental illness | 468 | 292 (176) | All above 50 | MINI | 43 (9.19) |
| Smith 2019 [138] | CS (9) | 2014–2015 | Mbarara | General populations | 1620 | 869 (751) | | DHSCL–(>1.75) | 460 (28.39) |
| Akimana 2019 [17] | CS (9) | 2015 | Kampala' | Children and adolescents with cancer | 352 | 112 (240) | 11.5±3.2; Range = 7–17 | MINI-KID | 91 (26) |
| Nampijja 2019 [139] | CS (9) | 2015 | Kampala | Postpartum women in Nsambya hospital | 300 | 300 (0) | 28±4.8; Range = 17–44 | MINI | 4 (1.3) |
| Cooper-Vince 2018 [140] | CS (9) | 2015 | Mbarara | Community-based study | 1603 | 898 (705) | | DHSCL–(>1.75) | 461 (28.76) |
| Kiene 2017 [141] | CS (8) | | | Individuals living with HIV | 325 | 165 (160) | Men = 34.93 ±10.59; Women = 32.21 ±8.93 | DHSCL–(>1.75) | 63 (19.4) |
| Kinyanda 2017b [142] | CS (9) | | Masaka, Wakiso | Individuals living with HIV | 899 | 705 (194) | | MINI | 126 (14.0) |
| Sohail 2019 [143] | CS (9) | 2015–2017 | Rakai | Rakai HIV community cohort | 333 | 160 (170) | 37± 9 | CES-D | 28 (8.4) |
| Raggio 2019 [144] | CS (8) | 2016 | Mbarara | Pregnant and non-pregnant women | 225 | 225 (0) | Median = 27; IQR = 9 | EPDS–(10) and DHSCL–(>1.75) | 26 (11.56) and 60 (26.67) |
| Kinyanda 2017a [145] | Cohort (9) | | Masaka, Wakiso | Individuals living with HIV | 1099 | 847 (252) | 35.1± 9.3 | MINI | 155 (14.1) |
| Akinbo 2016 [146] | CS (5) | | Bushenyi | Alcoholic youth | 204 | 71 (133) | 30.6% were 15–19 years old | Researchers designed questionnaire | 4 (2) |
| Cooper-Vince 2017 [147] | CS (9) | | Mbarara | Female household head | 257 | 257 (0) | 33.5±7.9 | DHSCL–(>1.75) | 133 (44) |
| Ashaba 2021 [148] | CS (9) | 2016–2017 | Mbarara | HIV positive adolescents | 224 | 131 (93) | 14.8±1.4. | MINI-KID | 37 (16) |
| Ashaba 2018 [149] | CS (9) | 2016–2017 | Mbarara | HIV positive adolescents | 224 | 131 (93) | 14.8±1.4. | MINI-KID | 37 (16) |
| Ortblad 2020 [150] | Prospective studies (9) | 2016–2019 | Kampala | Individuals living with HIV | 960 | | Median = 28, IQR = 8 | PHQ-9 | 416 (43.2) |
| Satinsky 2021 [151] | CS (9) | 2016–2018 | Mbarara | General populations | 1626 | 908 (718) | 17–26 = 409; 27–39 = 488; 40+ = 705 | DHSCL–(>1.75) and DSM V | 331 (20.36) and 159 (9.78) |
| Alinaitwe 2021 [152] | CS (9) | 2017 | Kampala | TB patients | 308 | 112 (188) | 36±10.8 | MINI | 73 (23.7) |
| Bapolisi 2020 [153] | CS (9) | 2017 | Isingiro | Refugees in Nakivale refugee camp | 387 | 219 (168) | 33.01±12.2 | MINI | 224 (58) |

(*Continued*)

**Table 1.** (Continued)

| First author, year of publication | Study design (JBI Checklist score) | Year of data collection | Districts | Study group | Sample size | Female (male) | Age (in years) | Tools used to access depression (cutoff) | Depression n (%) |
|---|---|---|---|---|---|---|---|---|---|
| Forry 2019 [154] | CS (9) | 2017 | Mbarara | Prison inmates in Mbarara Municipality | 414 | 25 (389) | Aged 22–35 years (60%) | MINI | 182 (44) |
| Seffren 2018 [155] | Cohort (9) | | Busia, Tororo | caregivers of HIV patients | 288 | 288 (0) | 33.5±5.8 | DHSCL–(>1.75) | 28 (9.7) |
| Kuteesa 2020 [156] | CS (9) | 2017 | Mukono | 15–24 years individuals in a fishing community | 1281 | 606 (675) | Range = 15–24 | PHQ-9 –(10) | 29 (2) |
| Mubangizi 2020 [54]– Preprint | CS (9) | 2017–2018 | Kampala | Adults with sickle cell disease at Mulago Sickle cell clinic | 255 | 161 (94) | Median = 21, IQR = 6 | SRQ-25 –(>5) | 174 (68.2) |
| Nabunya 2020 [157] | Longitudinal cluster randomized study (9) | 2017–2022 (baseline) | Five districts in southwestern Uganda | Adolescent girls in southern Uganda | 1260 | 1260 (0) | 15.4±; Range = 14–15 | BDI–(21) | 580 (46.03) |
| Arach 2020 [158] | CS (9) | 2018 | Lira | Postpartum women | 1789 | 1789 (0) | 25±7; Range = 12–47 | EPDS–(14) | 377 (21.1) |
| Logie 2020 [159] | CS (9) | 2018 | Kampala | Refugee and displaced youth aged 16–24 living in five informal settlements in Kampala | 445 | 333 (112) | | PHQ-9 –(10) | 297 (66.7) |
| Ssewanyana 2021 [32] | CS (7) | 2018 | Kampala | Patients with stomas | 15 | 11 (3) | Range = 18–60 | PHQ-9 –(5) | 13 (88) |
| Musinguzi 2018 [160] | CS (9) | | Masaka, Wakiso | Peri-urban Individuals living with HIV of Masaka and Entebbe | 201 | 160 (41) | 18–29 = 67; 30–34 = 39; 35–39 = 26; 40–49 = 48; 50+ = 21 | MINI | 62 (30.8) |
| Muliira 2019 [161] | CS (9) | | Kampala | Caregivers of cancer patients | 284 | 208 (76) | 36±13.8 | HADS–(8) | 137 (48.2) |
| Lukenge 2019 [52]–Thesis | CS (9) | 2018–2019 | Kampala | Women attending infertility clinic | 377 | 377 (0) | | PHQ-9 –(10) | 167 (44.3) |
| Misghinna 2020 [162] | CS (9) | 2018–2019 | Kampala | Refugees | 374 | 218 (156) | 37.1±12.1 | MINI | 174 (46.52) |
| Bahati 2021 [36]–Preprint | CS (5) | 2019 | Mbarara | Urban refugees in Mbarara municipality | 343 | 145 (198) | 28.8±11.0 | PHQ-9 –(10) | 329 (96) |
| Boduszek 2021 [163] | CS (9) | 2019 | | Primary and secondary school students aged 9–17 years | 11518 | 6035 (5483) | 14±1.95 | The 14-item Patient-Reported Outcomes Measurement Information System (PROMIS) Depression Short Form–(31) | 1224 (10.62) |
| Kabunga 2020 [37] | CS (6) | 2019 | Isingiro | Refugees in Nakivale camp | 146 | 76 (70) | 18–30 = 26; 31–44 = 42; 44–59 = 45; and 60+ = 27 | PHQ-9 –(10) | 119 (81) |

(*Continued*)

**Table 1.** (Continued)

| First author, year of publication | Study design (JBI Checklist score) | Year of data collection | Districts | Study group | Sample size | Female (male) | Age (in years) | Tools used to access depression (cutoff) | Depression n (%) |
|---|---|---|---|---|---|---|---|---|---|
| Kyohangirwe 2020 [164] | CS (9) | 2019 | Kampala | Adolescents | 281 | 176 (105) | 10–13 = 101; 14–17 = 180 | MINI-KID | 51 (18.2) |
| Olum 2020 [165] | CS (9) | 2019 | Kampala | Medical students at Makerere University | 331 | 133 (196) | 23.1±3.3 | PHQ-9 –(10) | 71 (21.5) |
| Mootz 2019 [166] | CS (9) | | Amuria, Katakwi, and Kumi | Girls and women war victims | 605 | 605 (0) | | DHSCL–(30) | 181 (30) |
| Ssebunnya 2019 [167] | CS (9) | | Kamuli | General populations | 1290 | 848 (442) | <25 = 281; 25–34 = 367; 35–44 = 302; >44 = 337 | PHQ-9 –(10) | 82 (6.4) |
| Kabunga 2021a [168] | CS (5) | 2020 | Isingiro | Refugees in Nakivale camp | 146 | 77 (69) | Majority of respondents 31.5% (n = 46) were aged between 44 and 59 | PHQ-9 –(10) | 66 (45.2) |
| Bongomin 2021 [33] | CS (9) | 2020 | Kampala | Patients with rheumatoid arthritis | 48 | 44 (4) | Median = 52; IQR = 17 | EQ-5D-5L anxiety/ depression dimension | 34 (70.8) |
| Kabunga 2021b [169] | CS (9) | 2020 | Lira | Out-of-school adolescents | 164 | 87 (77) | 10–13 = 13; 14–16 = 48; 17–19 = 103 | PHQ-9 –(10) | 55 (34) |
| Kizito 2020 [53]–Thesis | Cohort (9) | 2020 | Masaka | Postpartum mothers | 167 | 167 (0) | The majority were between 25–29 | EPDS | 58 (34.7) |
| Ouma 2021 [170] | CS (9) | 2020 | Gulu | Female sex workers | 300 | 300 (0) | 26.4±6.0 | MINI | 143 (47.7) |
| Najjuka 2020 [14] | CS (9) | 2020 | | University students | 321 | 123 (198) | 24.8±5.1 | Depression Anxiety and Stress Scale (DASS-21) | 259 (80.7) |
| Mahmud 2021 [171] | Cohort (9) | 2020 | Kyenjojo | General population | 1075 | | | PHQ-9 –(10) | 150 (14) |
| Nyundo 2020 [6] | CS (9) | | Iganga, Mayuge | Adolescents (10–19) | 598 | 286 (312) | 14.2±2.6 | the 6-Item Kutcher Adolescent Depression Scale (KADS-6) | 158 (26.5) |
| Kaggwa 2021 [16] | CS (9) | 2021 | Isingiro | Married/ cohabiting women | 153 | 153 (0) | 33.3±6.7. | PHQ-9 –(10) | 100 (65.4) |

Note: Table studies have been sorted according to the year of data collection. In those studies without the year of data collection, the year of submission to the journal was used to estimate the year of data collection. CS = Cross-sectional; DHSCL = Depression sub-section of the Hopkins Symptom Checklist; PHQ = Patient Health Questionnaire (PHQ); MINI = Mini-International Neuropsychiatric Interview; MINI-KID = Mini-International Neuropsychiatric Interview for Children and Adolescents; SRQ = Self-Reporting Questionnaire; CES-D = Center for Epidemiological Studies Depression Scale; EPDS = Edinburgh Postpartum Depression Scale; BDI = Beck Depression Inventory; CDI = Children Depression Inventory; HADS = Hospital Anxiety and Depression Scale; DSM = Diagnostic and Statistical Manual of Mental Disorders; CASI-5 = Child and Adolescent Symptom Inventory-5; DASS-21 = 21-item Depression Anxiety and Stress Scale; EQ-5D-5L = European Quality of Life Five Dimension-Five Level anxiety/depression dimension; KADS-6 = Kutcher Adolescent Depression Scale; PROMIS = Patient-Reported Outcomes Measurement Information System Depression Short Form

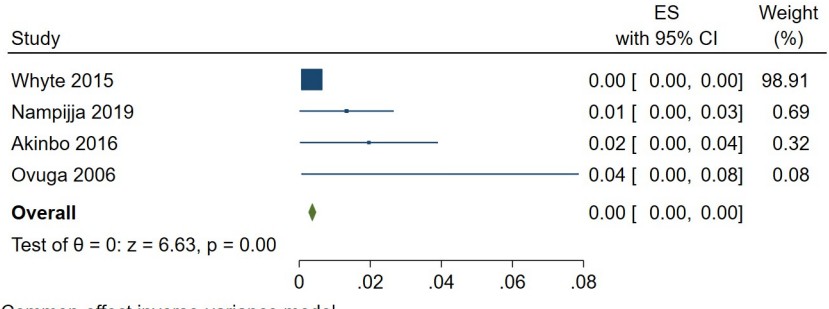

**Fig 2. Forest plot showing the pooled prevalence of depression in Uganda following sensitivity analysis.**

Questionnaire (PHQ) (*n* = 33, 26.0%) [16, 21, 32, 34, 36, 37, 52, 86, 88–90, 93, 98, 99, 101, 102, 112, 116–118, 120–123, 126–128, 131, 150, 156, 159, 165, 167–169, 171], (iii) Mini-International Neuropsychiatric Interview (MINI) (*n* = 24, 18.9%) [28, 29, 34, 59, 69, 70, 78, 94–96, 98, 103–105, 115, 119, 124, 134, 137, 139, 142, 145, 152–154, 160, 162, 170], (iv) Mini-International Neuropsychiatric Interview for Children and Adolescents (MINI-KID) (*n* = 6, 4.7%) [17, 58, 75–77, 107, 148, 149, 164], (v) Self-Reporting Questionnaire (SRQ) (*n* = 6, 4.7%) [31, 54, 57, 59, 83, 85] (vi) Center for Epidemiological Studies Depression Scale (CES-D) (n = 4, 3.1%) [15, 60, 72, 143], (vii) Edinburgh Postpartum Depression Scale (EPDS) (*n* = 4, 3.1%) [53, 97, 144, 158], (viii) Beck Depression Inventory (BDI) (*n* = 3, 2.4%) [56, 62, 157], (ix) Children Depression Inventory (CDI) (*n* = 2, 1.6%) [58, 114], (x) Hospital Anxiety and Depression Scale (HADS) standardized tool (*n* = 2, 1.6%) [106, 161], (xi) Diagnostic and Statistical Manual of Mental Disorders (DSM) (*n* = 2, 1.6%) [64, 151], (xii) Child and Adolescent Symptom Inventory-5 (CASI-5) (*n* = 1, 0.8%) [132], (xiii) European Quality of Life Five Dimension-Five Level (EQ-5D-5L) anxiety/depression dimension (*n* = 1, 0.8) [33], (xiv) the six-item Kutcher Adolescent Depression Scale (KADS-6) (*n* = 1, 0.8%) [6], (xv) the 14-item Patient-Reported Outcomes Measurement Information System (PROMIS) Depression Short Form (*n* = 1, 0.8%) [163], and (xvi) the Youth Self-Report (YSR) (*n* = 1) [51]. Two studies used a single self-report question to screen for depression [91, 130], and three studies developed their own tool to assess for depression [87, 129, 146]. One study used a clinician's diagnosis of depression [109]. The DASS-21 was also used in one study [14]. Six studies used more than one tool to screen and/or diagnose depression [34, 58, 59, 98, 144, 151].

## Prevalence of depression

A total of 27,989 (out of 123,859) individuals screened positive for depression. The pooled prevalence of depression was 30.2% (95% confidence interval [CI]: 26.7%-34.1%; $I^2$ = 99.80, $p<0.001$). The funnel plot was asymmetrical consistent with publication bias (S1 Fig). Despite the asymmetry, no studies were missing based on trim and fill analysis. The estimated slope from Egger's test was 6.12 (standard error [*SE*] = 0.598, $p<0.001$), suggesting publication bias due to small study effects. A sensitivity analysis was performed using the four studies within the funnel [56, 109, 139, 146]. The pooled prevalence of depression was 0.9% (95% CI: 0.1%-1.7%; $I^2$ = 37.82, $p$ = 0.021) (Fig 2). In univariate meta-regression analysis (to explore potential sources of heterogeneity), depression increased with the use of DASS-21 or SRQ-20 assessment tools and studies conducted during the COVID-19 pandemic. However, depression decreased with increased sample size and the number of females in the study.

**Table 2. Subgroup analysis of the prevalence of depression in Uganda.**

| Categories | Subgroups | Number of studies | Pooled prevalence (95% CI) | Q | $I^2$ | p-value | Group difference $\chi^2$ (p-value) |
|---|---|---|---|---|---|---|---|
| Pandemic | Pre-pandemic | 119 | 29.3 (25.5–33.0) | 23453.03 | 99.81 | <0.001 | **5.37 (0.021)** |
| | During the pandemic | 8 | 48.1 (32.6–63.6) | 319.21 | 96.74 | <0.001 | |
| Type of Study | Cross-sectional | 92 | 31.3 (26.7–35.9) | 17035.98 | 99.84 | <0.001 | 1.06 (0.588) |
| | Cohort | 32 | 29.0 (22.5–35.5) | 3616.12 | 99.43 | <0.001 | |
| | Case-control | 1 | 26.7 (19.0–34.5) | 0 | NA | NA | |
| Tools used | BDI | 3 | 20.5 (2.9–38.1) | 256.48 | 98.75 | <0.001 | **2205.32 (<0.001)** |
| | CASI-5 | 1 | 4.9 (3.7–6.1) | 0 | NA | NA | |
| | CDI | 2 | 36.6 (5.9–67.3) | 83.04 | 98.80 | <0.001 | |
| | CES-D | 4 | 31.1 (11.8–50.4) | 225.02 | 98.60 | <0.001 | |
| | Clinician diagnosis | 1 | 0.4 (0.2–0.5) | 0 | NA | NA | |
| | DASS-21 | 1 | 80.7 (70.9–90.5) | 0 | NA | NA | |
| | DHSCL | 34 | 33.7 (27.3–45.4) | 2022.12 | 99.25 | <0.001 | |
| | DSM V | 1 | 9.8 (8.3–11.3) | 0 | NA | NA | |
| | DSM IV | 1 | 39.8 (31.8–47.7) | 0 | NA | NA | |
| | EPDS | 4 | 27.1 (13.4–40.8) | 49.28 | 96.62 | <0.001 | |
| | EQ-5D-5L | 1 | 70.8 (47.0–94.6) | 0 | NA | NA | |
| | HADS | 2 | 37.3 (15.5–59.0) | 12.66 | 92.10 | <0.001 | |
| | KADS-6 | 1 | 26.4 (22.3–30.5) | 0 | NA | NA | |
| | MINI | 24 | 24.9 (18.1–31.8) | 1296.48 | 99.16 | <0.001 | |
| | MINI-KID | 6 | 17.6 (7.7–27.4) | 253.96 | 98.93 | <0.001 | |
| | Study designed tool | 3 | 31.5 (-0.9–63.9) | 450.27 | 99.47 | <0.001 | |
| | PHQ-2 | 2 | 36.1 (-4.77.7) | 174.21 | 99.43 | <0.001 | |
| | PHQ-9 | 31 | 31.0 (21.9–40.1) | 3475.21 | 99.78 | <0.001 | |
| | PROMIS | 1 | 10.6 (10.0–11.2) | 0 | NA | NA | |
| | SRQ-20 | 2 | 59.4 (29.6–89.1) | 13.75 | 92.72 | <0.001 | |
| | SRQ-25 | 4 | 33.4 (6.7–60.1) | 154.00 | 98.20 | <0.001 | |
| | Self-report | 2 | 41.9 (3.8–80.0) | 191.34 | 99.48 | <0.001 | |
| | YSR | 1 | 20.9 (17.1–24.8) | 0 | NA | NA | |
| Tool diagnostic | No | 94 | 33.2 (28.6–37.7) | 10489.46 | 99.68 | <0.001 | **8.03 (0.005)** |
| | Yes | 33 | 30.5 (26.7–34.2) | 3258.96 | 99.65 | <0.001 | |

DHSCL = Depression sub-section of the Hopkins Symptom Checklist; PHQ = Patient Health Questionnaire (PHQ); MINI = Mini-International Neuropsychiatric Interview; MINI-KID = Mini-International Neuropsychiatric Interview for Children and Adolescents; SRQ = Self-Reporting Questionnaire; CES-D = Center for Epidemiological Studies Depression Scale; EPDS = Edinburgh Postpartum Depression Scale; BDI = Beck Depression Inventory; CDI = Children Depression Inventory; HADS = Hospital Anxiety and Depression Scale; DSM = Diagnostic and Statistical Manual of Mental Disorders; CASI-5 = Child and Adolescent Symptom Inventory-5; DASS-21 = 21-item Depression Anxiety and Stress Scale; EQ-5D-5L = European Quality of Life Five Dimension-Five Level anxiety/depression dimension; KADS-6 = Kutcher Adolescent Depression Scale; PROMIS = Patient-Reported Outcomes Measurement Information System Depression Short Form

Due to the high heterogeneity, a subgroup analysis was performed. There were significant differences between the COVID-19 pandemic period (Q difference [QD] = 5.37, p<0.021), the assessment tool used for screening and/or diagnosing depression (QD = 2205.32, p<0.001), and the type of assessment tool used, i.e., diagnostic (DSM V, DSM VI, MINI, MINI-KID, and clinical diagnosis) or non-diagnostic (QD = 8.03, p = 0.005) (Table 2). The pooled prevalence of depression during the COVID-19 pandemic was higher than before the pandemic (48.1% vs. 29.3%). Fig 3 shows the forest plot of the prevalence of depression during the COVID-19 pandemic.

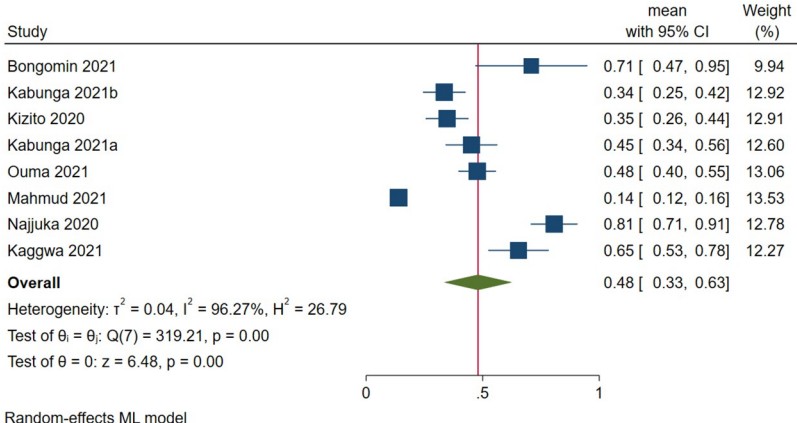

**Fig 3. Forest plot on the prevalence of depression during the COVID-19 pandemic.**

## Prevalence of depression in different study populations in Uganda

Depression was screened or diagnosed in the following study groups in Uganda: (i) individuals living with HIV (*n* = 43) [15, 34, 60, 64, 68, 70–73, 79, 81, 82, 85–91, 93–96, 98–105, 112–114, 117, 119, 123, 127, 128, 130, 132, 133, 136, 141–143, 145, 148–150, 160], (ii) females only (*n* = 25) [16, 31, 35, 52, 59, 73, 79, 97, 113, 116, 134, 139, 144, 158, 166, 170], including seven studies among pregnant or postpartum women [53, 59, 73, 97, 139, 144, 158], (iii) general population (*n* = 19) [55, 61–63, 78, 92, 110, 118, 120, 125, 140, 146, 151, 156, 167, 171], (iv) war victims (*n* = 12) [28, 29, 66, 67, 74–77, 80, 83, 84, 108, 111, 135, 166], (v) special patient groups (such as elderly, outpatients, diabetes mellitus, post-tuberculosis lung diseases, physically ill, post-stroke, cancer, tuberculosis patients, sickle cell disease, patients with stoma, rheumatoid arthritis) (*n* = 12) [17, 21, 31–33, 54, 57, 109, 115, 122, 124, 131, 152], (vi) children and adolescents (*n* = 10) [6, 17, 51, 58, 75–77, 84, 85, 107, 114, 129, 146, 157, 163, 164, 169], (vii) refugees (*n* = 8) [35–37, 116, 153, 159, 162, 168], (viii) caregivers of patients (*n* = 6) [106, 132, 134, 137, 155, 161], (ix) university students (*n* = 4) [14, 56, 65, 165], (x) prisoners (*n* = 1) [154], and males only (*n* = 1) [121] (S1 Table). The pooled prevalence of depression across the different study groups was highest among refugees (67.6%) and lowest among caregivers of patients (18.5%) (Fig 4).

## Depression among refugees in Uganda

A total of 2,538 refugees were assessed for depression in Uganda, and 1,652 screened positive for depression in eight studies. The prevalence of depression ranged between 45.2% [168] and 96% [36]. The pooled prevalence of depression was 67.6% (95 CI: 53.7%-81.5%; $I^2$ = 94.82, *p*<0.001) (Fig 5). The estimated slope from Egger's test was 6.67 (*SE* = 3.037, *p*<0.0281), suggesting publication bias due to small study effects. The funnel plot showed publication bias on visual inspection. A sensitivity analysis was performed using studies within the funnel [37, 159], and the pooled prevalence of depression was 72.7% (95 CI: 58.5%-87.0%; $I^2$ = 67.51, *p*<0.001). Based on meta-regression, the prevalence decreased with the increase in mean age (*β* = -0.07, *SE* = 0.02, *p*<0.001). Other factors were not significant.

**Depression among studies involving only females in Uganda.** A total of 4,222 out of 11476 females had depression in 25 female-only studies. The prevalence of depression ranged between 6.1% [59] and 92.0% [35]. The pooled prevalence of depression was 38.2% (95 CI:

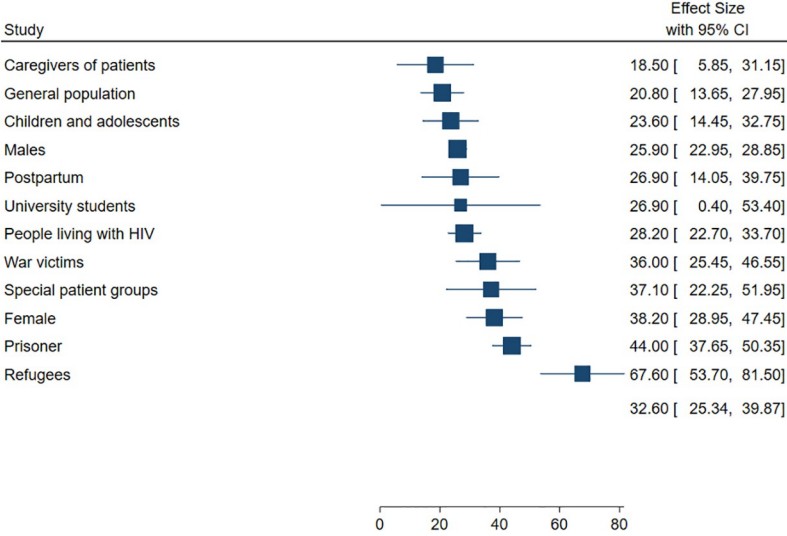

**Fig 4. Prevalence of depression in different study groups in Uganda.**

29.0%-47.5%; $I^2$ = 99.16, $p$<0.001) (Fig 6). The funnel plot showed asymmetrical distribution, therefore showing publication bias. The estimated slope from Egger's test was 8.65 ($SE$ = 0.1.428, $p$<0.001), suggesting publication bias due to small study effects. Following meta-regression, no factor significantly affected the prevalence of depression among females in Uganda.

**Depression among postpartum or pregnant women in Uganda.** Out of the 6296 postpartum or pregnant women in Uganda, 2028 had depression in seven studies. The prevalence of depression ranged between 1.3% [139] and 45.2% [83]. The pooled prevalence of depression among postpartum or pregnant women was 26.9% (95 CI: 13.6%-40.3%; $I^2$ = 99.44, $p$<0.001) (Fig 7). The estimated slope from Egger's test was 6.76 ($SE$ = 3.694, $p$<0.067), suggesting no publication bias due to small study effects. Based on meta-regression, the prevalence of depression among postpartum women statistically significantly decreased with use of MINI ($\beta$ = -0.35, $SE$ = 0.11, $p$ = 0.002) and increased when the study design was cohort ($\beta$ = 0.22, $SE$ = 0.11, $p$ = 0.045).

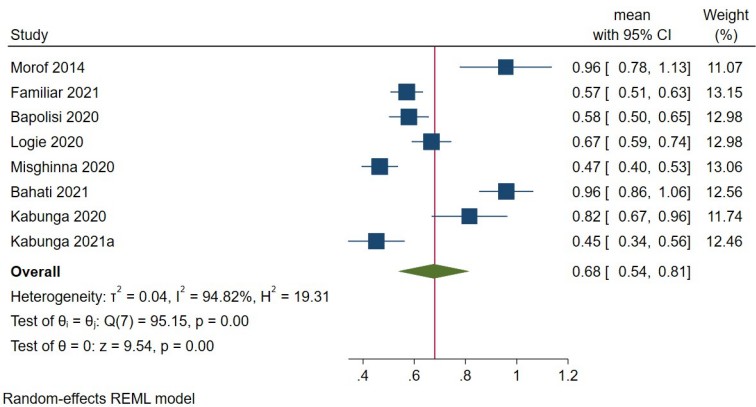

**Fig 5. Forest plot on the prevalence of depression among refugees in Uganda.**

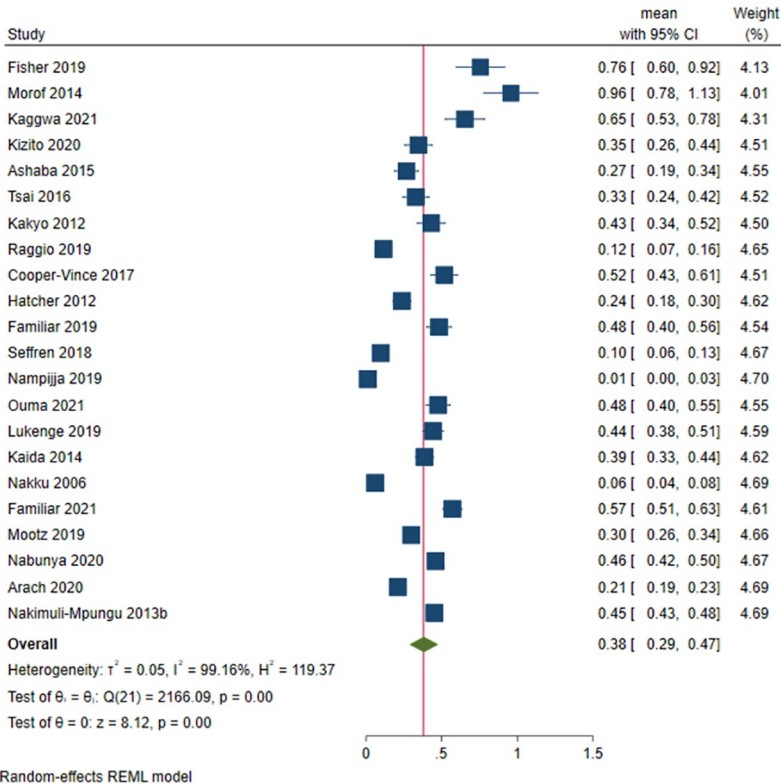

**Fig 6. Forest plot on the prevalence of depression among females in Uganda.**

**Depression among special patient groups in Uganda.** The prevalence of depression ranged from 0.4% among outpatients in northern Uganda [109] and 88% among patients with stomas [32]. The prevalence among those who (i) were elderly was 18% [57], (ii) had tuberculosis was 23.7% [152], (iii) had post-tuberculosis lung diseases was 24% [122], (iv) had cancer was 26% [17], (v) with post-stroke was 31.5% [131], (vi) were physically ill was 33.7% [124], (vii) had diabetes mellitus was 34.8% [115], (viii) had sickle cell disease was 68.2% [54], and (ix) had rheumatoid arthritis was 70.8% [33]. The pooled prevalence of depression from the 14,405 special patient groups (of whom 855 had depression) in 12 studies was 37.1% (95 CI: 22.3%-52.0%; $I^2$ = 99.55, $p$<0.001) (Fig 8). The estimated slope from Egger's test was 3.82 ($SE$ = 1.124, $p$<0.001), suggesting publication bias due to small study effects. Due to the significant heterogeneity, a sensitivity analysis was performed with studies within the funnel, and [115, 122, 124, 131], the pooled prevalence of depression in these studies was 33.8% (95 CI: 29.8%-37.9%; $I^2$ = 0.03, $p$<0.001). At meta-regression, the prevalence of depression among special patient groups statistically significantly increased with increase in the mean age ($\beta$ = 0.01, $SE$ = 0.004, $p$ = 0.009) and use of SRQ-20 to assess depression ($\beta$ = 0.75, $SE$ = 0.30, $p$ = 0.013).

**Depression among war victims in Uganda.** A total of 6583 (out of 19255) war victims had depression in 12 studies. The prevalence of depression ranged between 7.6% [84] and 71% [67]. The pooled prevalence of depression was 36.0% (95 CI: 25.5%-46.6%; $I^2$ = 99.50, $p$<0.001) (Fig 9). The estimated slope from Egger's test was 5.24 ($SE$ = 2.109, $p$ = 0.013), suggesting publication bias due to small study effects. Only one study was within the funnel [75].

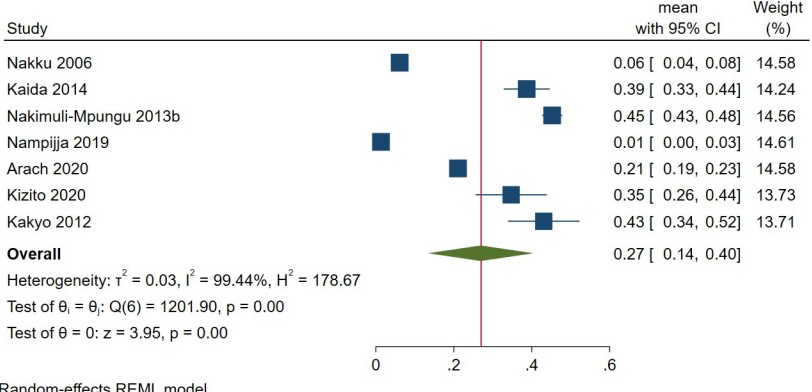

**Fig 7. Forest plot on the prevalence of depression among postpartum or pregnant females in Uganda.**

At meta-regression, no factor significantly affected the prevalence of depression among war victims in Uganda.

**Depression among individuals living with HIV in Uganda.** A total of 7704 (out of 26255) individuals living with HIV had depression in 43 studies. The prevalence of depression ranged between 5% [119] and 84% [34]. The pooled prevalence of depression was 28.2% (95 CI: 22.7%-33.7%; $I^2$ = 99.16, $p<0.001$) (Fig 10).

The estimated slope from Egger's test was 5.72 ($SE$ = 0.1406, $p<0.001$), suggesting publication bias due to small study effects. At meta-regression, no factor statistically significantly affected the prevalence of depression among HIV patients in Uganda.

**Depression among university students in Uganda.** A total of 517 (out of 1982) university students had depression in five studies. The prevalence of depression ranged between 0.4% [56] and 80.7% [14]. The pooled prevalence of depression among university students was 26.9% (95 CI: 0.4%-53.4%; $I^2$ = 99.48, $p<0.001$) (Fig 11). The estimated slope from Egger's test was 19.85 ($SE$ = 4.566, $p<0.001$), suggesting publication bias due to small study effects. The

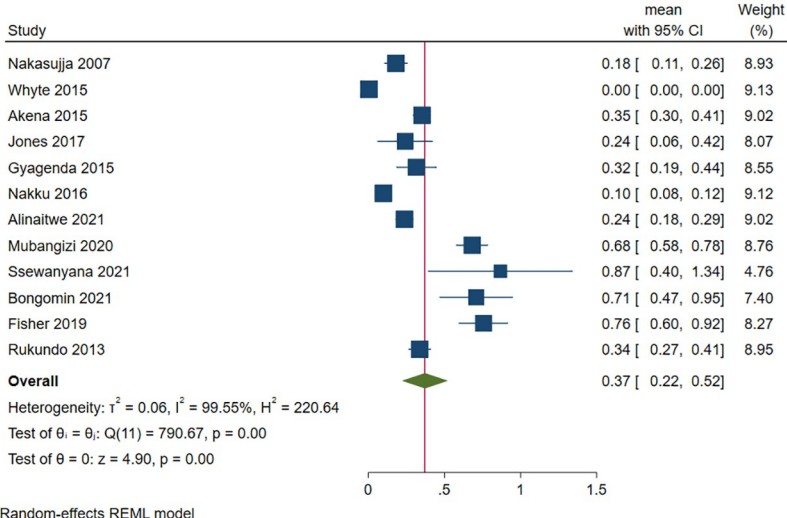

**Fig 8. Forest plot on the prevalence of depression among special patient groups in Uganda.**

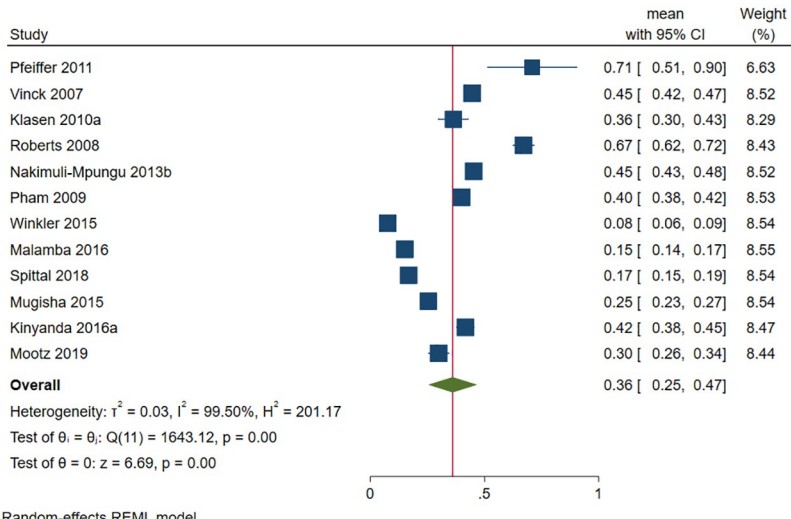

**Fig 9. Forest plot on the prevalence of depression among war victims in Uganda.**

pooled prevalence from the studies within the funnel during sensitivity analysis was 14.9% (95 CI: 12.7%-17.0%; $I^2$ = 0.04, $p$<0.001). The prevalence of depression among students was significantly higher when the DASS-21 was used to screen for depression ($\beta$ = 0.71, $SE$ = 0.10, $p$<0.001) and when the study was conducted during the COVID-19 pandemic ($\beta$ = 0.67, $SE$ = 0.09, $p$<0.001).

**Depression among children and adolescents in Uganda.** A total of 2535 (out of 17072) children and adolescents in Uganda screened positive for depression in 10 studies. The prevalence of depression among children and adolescents ranged from 2.9% [58] to 46.03% [157]. The pooled prevalence of depression among children and adolescents was 23.6% (95 CI: 14.5%-32.8%; $I^2$ = 99.55, $p$<0.001) (Fig 12).

The estimated slope from Egger's test was 5.27 ($SE$ = 2.008, $p$ = 0.009), suggesting publication bias due to small study effects. Only two studies were within the funnel [6, 51], and sensitivity analysis based on these studies had a pooled prevalence of 23.6% (95 CI: 18.3%-29.0%; $I^2$ = 72.52, $p$<0.001). The prevalence of depression was significantly lower when the following assessment tools were used: (i) MINI-KID ($\beta$ = -0.37, $SE$ = 0.09, $p$<0.001), (ii) PROMIS ($\beta$ = -0.35, $SE$ = 0.11, $p$ = 0.002), and (iii) YSR ($\beta$ = -0.25, $SE$ = 0.12, $p$ = 0.032).

**Depression among caregivers of patients in Uganda.** Different types of caregivers were included in this review and they included caregivers for the following patients: individuals living with (i) HIV ($n$ = 3) [112, 132, 155], cancer ($n$ = 2) [106, 161], and (iii) mental health illness ($n$ = 1) [137]. A total of 2189 (out of 14727) caregivers had depression. The pooled prevalence of depression was 18.5% (95 CI: 5.9%-31.2%; $I^2$ = 99.62, $p$<0.001) (Fig 13). The estimated slope from Egger's test was 6.88 ($SE$ = 2.710, $p$<0.011), suggesting publication bias due to small study effects. Only one study was inside the funnel [106]. At meta-regression, no factor statistically significantly affected the prevalence of depression among caregivers of patients in Uganda.

**Depression among the general population.** A total of 4,250 (out of 21,347) members of the general population screened positive for depression in 19 studies. The prevalence of depression ranged between 2.0% among individuals in a fishing community [156] and 68% among national humanitarian aid workers [92]. The pooled prevalence of depression was

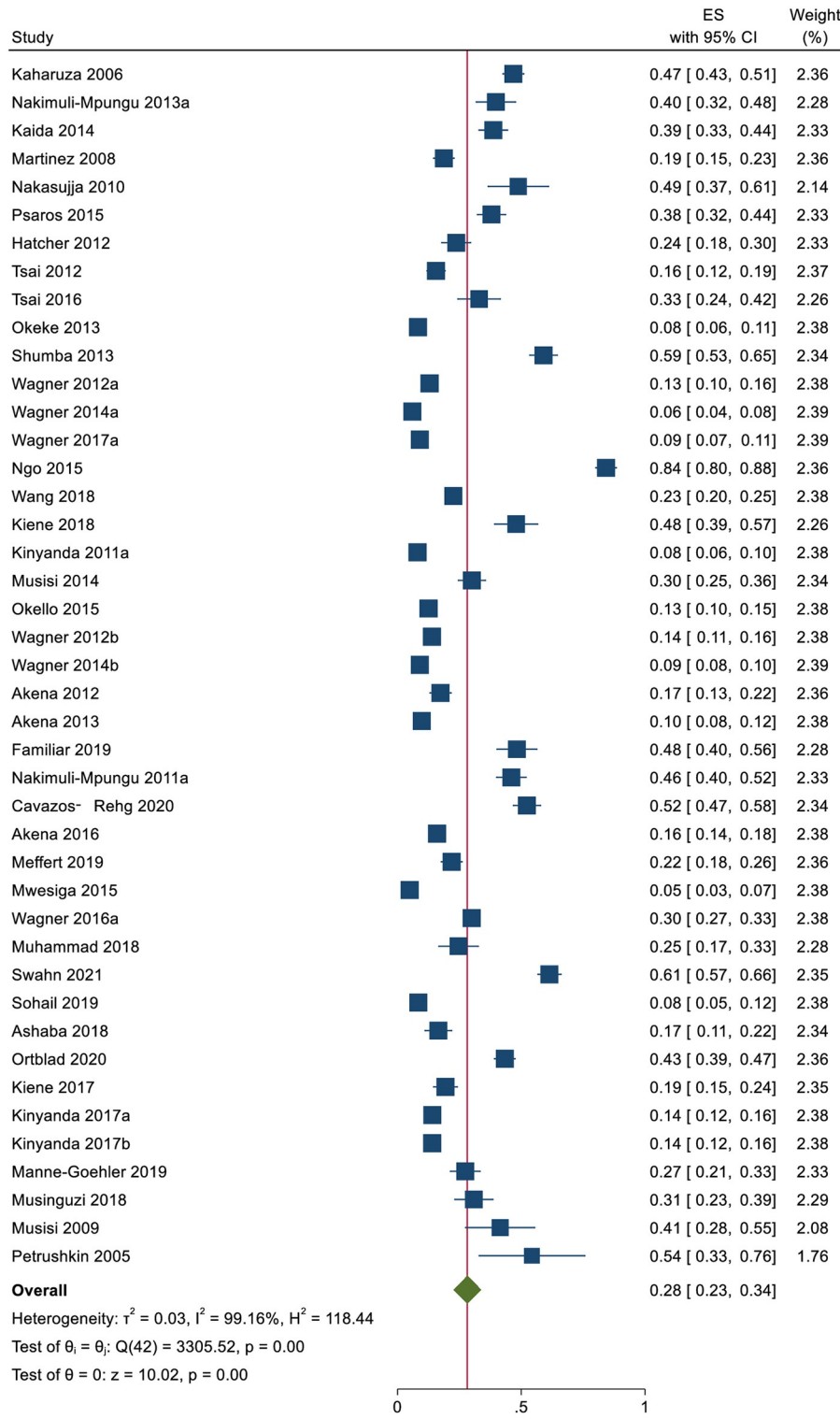

**Fig 10. Forest plot on the prevalence of depression among individuals living with HIV in Uganda.**

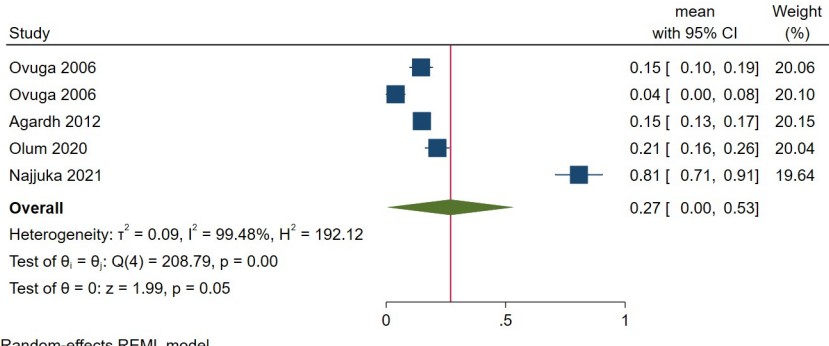

**Fig 11. Forest plot on the prevalence of depression among university students in Uganda.**

20.8% (95 CI: 13.6%-27.9%; $I^2$ = 99.61, $p$<0.001) (Fig 14). The estimated slope from Egger's test was 10.91 ($SE$ = 1.889, $p$<0.001), suggesting publication bias due to small study effects. At meta-regression, no factor statistically significantly affected the prevalence of depression among the general population in Uganda.

## Discussion

The present systematic review and meta-analysis pooling data of close to 124,000 Ugandans collected between 2000 and 2021 showed that approximately one in three individuals had depression. This finding is much higher than the global depression rate of 3.8% [2]. This large difference may be because the majority of the studies included in this review involved study populations that are at higher risk of developing depression, such as refugees, war victims, individuals living with HIV, and caregivers of patients, among others [172–175]. However, the prevalence of depression in Uganda was slightly higher than 27% from a previous systematic review and meta-analysis of the prevalence of depression among outpatients [13].

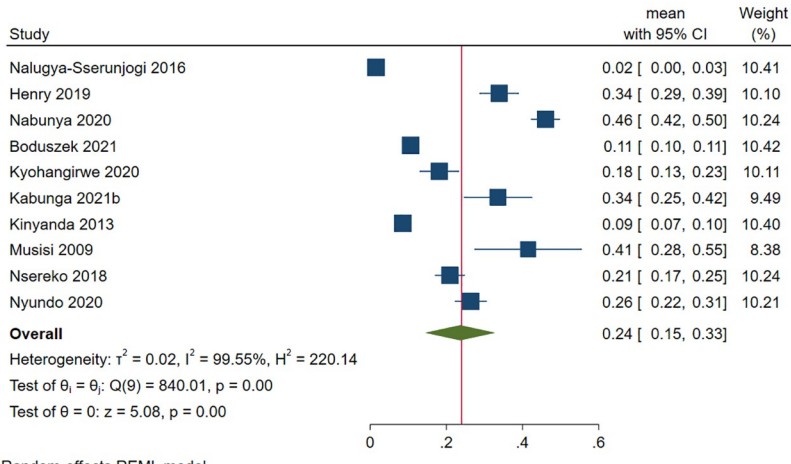

**Fig 12. Forest plot on the prevalence of depression among children and adolescents in Uganda.**

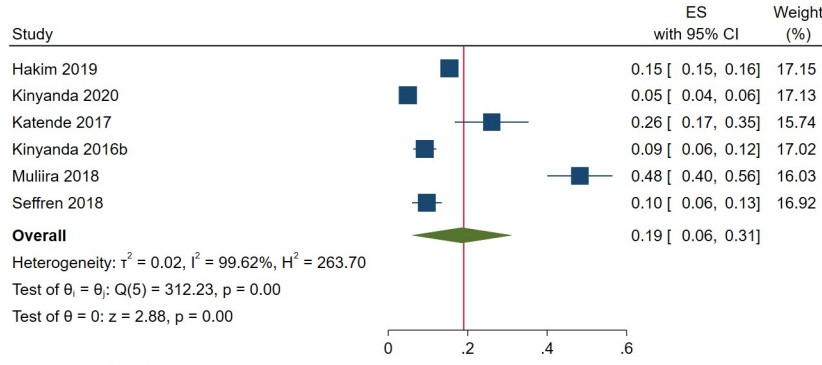

**Fig 13. Forest plot on the prevalence of depression among caregivers of selected patient groups in Uganda.**

The prevalence of depression was also higher than previously obtained pooled prevalence rates of depression in Uganda (21.2% among adults and 20.2% among children for studies published between 2010 and 2018 [12]). Since all the previous review studies are included in this study, the difference between the pooled prevalence of depression between the present study and the previous reviews may be due to the effect of the COVID-19 pandemic that led to increased levels of depression [176]. This was clearly indicated by the subgroup analysis, which showed a higher difference between the pre-pandemic pooled prevalence of depression and that during the pandemic. The present systematic review had a different prevalence than the former studies because it included more studies which could have resulted in the pooled prevalence rate being closer to "the true value" of the prevalence of depression in Uganda.

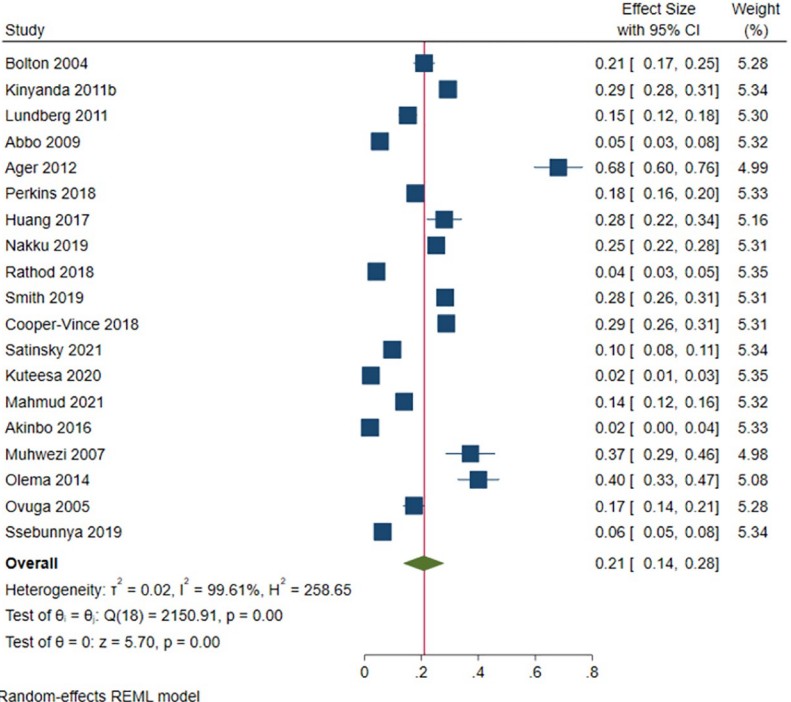

**Fig 14. Forest plot on the prevalence of depression among the general population in Uganda.**

The prevalence of depression in the different study groups was highest among refugees (67.6%) compared to other groups. This prevalence was over twice as high as a previously reported prevalence of depression among refugees and asylum seekers (31.5%) [172]. Uganda, the world's fourth largest refugee hosting country, has been host to refugees from Congo, South Sudan, Rwanda, Burundi, Somalia, and Ethiopia, among other countries [25]. The high prevalence of depression may be due to refugees leaving their countries to come to a low-income country that is also affected by multiple health, social, and financial struggles, leaving many refugees with depression or worsening their psychological states [172]. The higher prevalence of depression among refugees compared to other studied groups may be because these refugees, on top of their struggles and settling into a new environment, are also affected by the challenges of the country to which other groups are used to.

Uganda has also been affected by civil wars, especially in the northern part of the country. The prevalence of depression among the war victims in Uganda (36.0%) was higher than the 27% global estimate from a systematic review and meta-analysis of war victims [173]. The psychological impact of civil war, refugees, and wars in the neighboring countries on the victims and the workers may be the high prevalence of depression among the national humanitarian workers compared to the rest of the general population.

Despite the declining prevalence of HIV among Ugandans [177], many individuals were affected by the mental and psychological impacts of HIV, such as depression [38, 178]. The prevalence of depression among individuals living with HIV in Uganda (28.2%) in the present study was lower than the global prevalence of 31% [179]. The lower prevalence may be attributed to the efforts made by many researchers to understand and reduce the burden of depression, as evidenced by the high number of studies regarding depression in the present review.

The prevalence was also lower than the previous prevalence of depression among Ugandans with HIV (30.88%) that involved studies published before 2018 [38]. This difference may be attributed to a few studies being included in the previous review (n = 10) [38]. In Uganda, depression among individuals living with HIV has been studied widely and has been assessed as various risk factors such as depression among caregivers of individuals living with HIV [112, 132, 155]. Based on the present review, caregivers of patients have less depression compared to the patients and other study groups. However, they play an integral role in patient care. The prevalence of depression among special groups of caregivers, such as cancer patient caregivers, was higher (42.3%) [180] compared to the pooled prevalence in the present review. This difference may be attributed to the Ugandan culture, where the caregiving role is shared among all family members and creates family support for the affected caregivers, helping prevent depression [161, 181, 182].

Being female was highly represented in the review, with a total of 25 studies being carried out among female-only studies compared to only one male-only study. This possibly shows neglect of male gender mental health by researchers. Future research should include more studies among males, so that true estimates of the burden of depression can be determined and evidence-based interventions can be designed. Depression among children and adolescents has also been studied more than studies of male adults. The prevalence of depression among children and adolescents (23.6%) was higher than 20.2% among children in Uganda for papers published between 2010 and 2018 [12].

Despite having no missing studies imputed into the overall prevalence, the heterogeneity was high. Following sensitivity analysis, the prevalence of depression in Uganda was 0.9%—a prevalence lower than the estimated global prevalence of 3.8% [2]. Based on the various analyses, the main sources of heterogeneity were (i) the COVID-19 pandemic, where the prevalence of depression was significantly higher than in the period before the pandemic as reported by various researchers and meta-analyses [176, 183]; and (ii) the tools used in screening/

diagnosing depression with the DASS-21 detecting significantly higher prevalence rates of depression compared to other study tools. The significant difference may be due to the tool being used during the early stages of the COVID-19 pandemic [14] when many of the individuals were experiencing severe depression due to various stressors [176, 183]. The difference in the reported prevalence of depression could be due to various studies using different assessment tools with different psychometric properties regarding depression. Also, some tools were diagnostic, such as the DSM criteria, while others were screening tools, such as the DASS-21.

## Limitations and recommendations

When interpreting these results, the following limitations need to be considered. First, despite only 16% of the 127 papers not having a total score of nine on the JBI Checklist and the use of random effect models, there were significantly high levels of heterogeneity due to the depression assessment tools and the period of study. Future researchers should conduct reviews of studies with fewer variations, especially in relation to the tools used to assess depression. However, for better quality and to increase reliability in future meta-analyses, future researchers should continue using the commonly used tools such as PHQ-9, DHSCL, and MINI. Also, the classification of the different study groups in the present study may have caused heterogeneity in the included studies, for example, among the general population. Second, some of the included studies were prone to recall biases since all their data were based on self-report. Third, despite data from various regions and districts in Uganda being presented, a large majority of the country was still not represented. This suggests more research regarding depression in other parts of this multicultural and multilingual country should be conducted and/or a nationally representative survey study [18]. Moreover, despite conducting a detailed literature search, some of the common databases (e.g., *EMBASE*, *CINAHL*) and journals that publish papers on mental health illness were not included. Therefore, some studies could have been missed. Also, the search strategy did not include some of the common terms associated with depression, such as mental health, psychological disorder/problem, and mood. It is recommended that future studies include sources for unpublished data to generalize the findings better.

While the meta-analysis was comprehensive and provided a broader picture of the prevalence of depression in various populations, it is still difficult to generalize the results because the prevalence of depression in Uganda in many regions was not represented, and different populations' generalizations or groupings were subjective (e.g., humanitarian workers). Future studies within these populations and across wider regions in the country would be helpful in implementing treatments according to targeted needs (socioeconomic, cultural, refugee-status, etc.).

## Conclusion

In the present meta-analysis, the synthesized data showed that approximately one in three individuals in Uganda has depression, which was highest among refugees and other special populations. Interventions for active screening, diagnosis, and management of depression among the general population and special populations and cohorts are recommended.

## Supporting information

**S1 Fig. Funnel plot for the included studies about depression in Uganda.**
(TIF)

**S2 Fig.**
(TIF)

**S1 Table. Prevalence of depression in study populations in Uganda.**
(DOCX)

## Acknowledgments

We acknowledge the support and guidance provided by the Librarians at Mbarara University of Science and technology in developing the search strand strings and literature retrieval for the selected papers.

## Author Contributions

**Conceptualization:** Mark Mohan Kaggwa, Sarah Maria Najjuka.

**Data curation:** Mark Mohan Kaggwa, Sarah Maria Najjuka.

**Formal analysis:** Mark Mohan Kaggwa, Sarah Maria Najjuka, Mohammed A. Mamun.

**Investigation:** Mark Mohan Kaggwa, Sarah Maria Najjuka, Felix Bongomin, Mohammed A. Mamun, Mark D. Griffiths.

**Methodology:** Mark Mohan Kaggwa, Sarah Maria Najjuka, Felix Bongomin, Mohammed A. Mamun, Mark D. Griffiths.

**Project administration:** Mark Mohan Kaggwa, Sarah Maria Najjuka.

**Resources:** Mark Mohan Kaggwa.

**Software:** Mark Mohan Kaggwa.

**Supervision:** Mark Mohan Kaggwa, Felix Bongomin, Mark D. Griffiths.

**Validation:** Mark Mohan Kaggwa, Sarah Maria Najjuka, Felix Bongomin, Mark D. Griffiths.

**Visualization:** Mark Mohan Kaggwa, Sarah Maria Najjuka, Felix Bongomin, Mohammed A. Mamun, Mark D. Griffiths.

**Writing – original draft:** Mark Mohan Kaggwa, Sarah Maria Najjuka.

**Writing – review & editing:** Mark Mohan Kaggwa, Sarah Maria Najjuka, Felix Bongomin, Mohammed A. Mamun, Mark D. Griffiths.

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
