## [Decision Letter · Decision Letter 0]

19 Jul 2022

PONE-D-22-10716Prevalence of depression in Uganda. A systematic review and meta-analysisPLOS ONE

Dear Dr. Kaggwa,

Thank you for submitting your manuscript to PLOS ONE. After careful consideration, we feel that it has merit but does not fully meet PLOS ONE’s publication criteria as it currently stands. Therefore, we invite you to submit a revised version of the manuscript that addresses the points raised during the review process.

We look forward to receiving your revised manuscript.

Kind regards,

Muhammed Elhadi, MBBCh

Academic Editor

PLOS ONE

Journal Requirements:

4. We note that Figure 2 in your submission contain Map images which may be copyrighted. All PLOS content is published under the Creative Commons Attribution License (CC BY 4.0), which means that the manuscript, images, and Supporting Information files will be freely available online, and any third party is permitted to access, download, copy, distribute, and use these materials in any way, even commercially, with proper attribution. For these reasons, we cannot publish previously copyrighted maps or satellite images created using proprietary data, such as Google software (Google Maps, Street View, and Earth). For more information, see our copyright guidelines: http://journals.plos.org/plosone/s/licenses-and-copyright.

Natural Earth (public domain): http://www.naturalearthdata.com

5. Please upload a new copy of Figure 11 as the detail is not clear. Please follow the link for more information: https://blogs.plos.org/plos/2019/06/looking-good-tips-for-creating-your-plos-figures-graphics/" https://blogs.plos.org/plos/2019/06/looking-good-tips-for-creating-your-plos-figures-graphics/

Additional Editor Comments :

We request that a point-by-point response letter accompanies your revised manuscript. This letter must provide a detailed response to each reviewer/editorial point raised, describing what amendments have been made to the manuscript text and where these can be found (e.g. Methods section, line 12, page 5). If you disagree with any comments raised, please provide a detailed rebuttal to help explain and justify your decision.

The way of presenting data is needing major and detailed revision , more clarity on the analysis process can be addressed in a revision.

There are many formatting issues that may need to be addressed (many spacing issues). The Figures seem to also need to be higher quality.

The reporting of results and discussion requires revisions to be more succinct and concise. Some reviewers were suggesting rejection or major revision.

Additional comments are described in the feedback to the authors.

Reviewers' comments:

Reviewer's Responses to Questions

**Comments to the Author**

1. Is the manuscript technically sound, and do the data support the conclusions?

Reviewer #1: Partly

Reviewer #2: Yes

Reviewer #3: Yes

Reviewer #4: Yes

Reviewer #5: Yes

Reviewer #6: Partly

Reviewer #7: Yes

Reviewer #8: Yes

Reviewer #9: Yes

Reviewer #10: Yes

Reviewer #11: No

2. Has the statistical analysis been performed appropriately and rigorously? 

Reviewer #1: Yes

Reviewer #2: Yes

Reviewer #3: Yes

Reviewer #4: Yes

Reviewer #5: I Don't Know

Reviewer #6: I Don't Know

Reviewer #7: Yes

Reviewer #8: Yes

Reviewer #9: Yes

Reviewer #10: Yes

Reviewer #11: No

3. Have the authors made all data underlying the findings in their manuscript fully available?

Reviewer #1: Yes

Reviewer #2: Yes

Reviewer #3: Yes

Reviewer #4: Yes

Reviewer #5: Yes

Reviewer #6: Yes

Reviewer #7: No

Reviewer #8: Yes

Reviewer #9: Yes

Reviewer #10: Yes

Reviewer #11: Yes

4. Is the manuscript presented in an intelligible fashion and written in standard English?

Reviewer #1: Yes

Reviewer #2: Yes

Reviewer #3: Yes

Reviewer #4: Yes

Reviewer #5: Yes

Reviewer #6: Yes

Reviewer #7: Yes

Reviewer #8: Yes

Reviewer #9: Yes

Reviewer #10: Yes

Reviewer #11: No

5. Review Comments to the Author

Reviewer #1: This study aimed to determine the pooled prevalence of depression in Uganda and determine the prevalence of depression among various study populations in the country. The strength of this study was to conduct analysis using a rigorous method of systematic reviews. However, there were some concerns in this study.

First, in the Abstract, the authors had better adhere to PRISMA 2020 for Abstract. For example, they had better describe the following information in the abstract: inclusion criteria and exclusion criteria for the review, the date when each database was last searched, the methods used to assess the risk of bias in the included studies, the methods used to present and synthesize results, the number of participants for the prevalence of depression in refugees, war victims, individuals living with HIV, postpartum or pregnant mothers, university students, children and adolescents, caregivers of patients, a brief summary of the limitations of the evidence included in the review (e.g. study risk of bias, inconsistency and imprecision), the primary source of funding for the review, the register name and registration number.

Second, in Figure 1, they had better report the correct number for records. For example, the sum of records for databases is not n=513 whereas they reported the sum of records for databases is n=513. Moreover, they reported that they excluded 103 records from 216 records, then 121 records remained.

Third, they had better describe the results of each nine-item for JBI checklist in results. They had better describe the discussion for the results of each item for the JBI checklist in the Discussion.

Reviewer #2: Dear authors,

Thanks for submitting your manuscript “Prevalence of depression in Uganda. A systematic review and meta-analysis” to be considered for publication in PlosONE.

Very well written and interesting paper, methodologically strong and relevant for the journal and the field of global mental health. There are only minor inaccuracies which I think should be addressed before publication. Please, refer to my detailed comments below.

Regards

A.

Table 1. May I ask why the articles have not been reported in alphabetical order, or what was the sorting rule used? It looks like year (oldest -> newest), but year of data collection is missing for some studies. I am not asking to change it, just to clarify, maybe in Table caption or as a note?

Moreover, I think it would be good to add “(years)” to Age, as you did for the other variable and, maybe, instead of n you could report % Female?

At page 25, L176, you reported p<0.001, is this in relation to heterogeneity? This also applies to page 26, L206 and on all other occasions. As before, maybe it would be good to state it clearly in the text, e.g., by using a semi-colon before I2?

Please, correct “we” to “were” at L176.

There is something weird in the PDF at P26, L182, could you make sure, in your doc version, that all references in relation to “Depression sub-section of 182 the Hopkins Symptom Checklist (DHSCL)..” have been included?

Considering you detected publication bias, would you consider using trim and fill analysis to estimate the number of missing studies? Good that you carried out a sensitivity analysis, at L209.

P28, could you please consider adding a statement of the results presented in Table 2, in relation to Tools used?

Could you please consider improving readability in sentence “The prevalence among the elderly [140], 288 tuberculosis patients [111], post tuberculosis lung diseases [91], cancer [11], post-stroke 289 [86], physically ill [117], diabetes mellitus [119], sickle cell disease [43], and rheumatoid 290 arthritis [27] were 18%, 23.7%, 24%, 26%, 31.5%, 33.7%, 34.8%, 68.2%, and 70.8%, 291 respectively.”? Maybe adding each percentage close to the group label?

Previous systematic review and MA (166) is only cited in the discussion, but I think it would be good to mention, in the introduction, how yours is different from previous SR/MA already published on the topic.

I have particularly appreciated the discussion on factors potentially leading to increased depression rates in Uganda, compared to other countries.

Reviewer #3: Recently, Opio JN et al. performed a meta-analysis to determine the prevalence of mental disorders in Uganda, including major depressive disorder. However, to date, the present work is the first meta-analysis specifically addressing the prevalence of depression in Uganda, with a very large sample size and across several special populations (caregivers, individuals living with HIV or other chronic disease, etc.). The statistical analysis was correctly performed, as well as the risk of bias and the quality of the included studies were correctly assessed.

Reviewer #4: This systematic review reports the prevalence of depression in Uganda. This review is written according to the PRISM guidelines. The authors know the limitations on the judge of depression. It is important to be interpreted.

Reviewer #5: The authors aimed to assess the pooled prevalence of depression and the prevalence of depression among various study populations in Uganda. The study gives updates on the prevalence of depression in the country and the distribution of depression by population and region. The manuscript was well-written. There are some issues on eligibility criteria, discussion of the high prevalence of depression in this study, and limitations that need to be addressed. Below are more specific comments by section:

2.2 Inclusion criteria and exclusion criteria:

- If there were other pre-specified of eligibility criteria, including language (e.g. English, Luganda), definition of depression (e.g. depressed mood, depressive disorder), assessment tools for depression (e.g. type, cutoff point), minimum sample size, study quality (e.g. JBI score of 4 or more), they should be mentioned in this section.

- Having information on prevalence of depression should be mentioned as one of the inclusion criteria.

4. Discussion:

(Line 372-387) The finding that the pooled prevalence of depression in Uganda from the present study is higher than that from the previous study might be due to the difference in assessment tools for depression between the two studies, besides the effects of COVID-19 pandemic. Specifically, the previous review by Opio used more strict criteria for depressive disorder. Most of the included studies in Opio’s review used M.I.N.I., a structural interview, to assess depressive disorder.

4.1 Limitations:

- (Lines 433-435) How “high levels of heterogeneity due to the depression assessment tools and the period of the study” affect the results/implementation of the results. This should be explained.

- (Lines 436-437) “Second, the included studies were all prone to recall biases, since all the data were self-report.” I am not sure if this statement is correct. For example, M.I.N.I. is a structured interview, not a self-report measure.

- (Lines 437-439) How “a large majority of the country was still not represented” affect the results/implementation of the results. This should be explained.

- Some relevant literature in other languages (e.g. Luganda) or from other databases (e.g. EMBASE, CINAHL) might be missed. This should be mentioned as a limitation.

- Some relevant terms for depression, such as mental health, psychological disorder/problem, and mood, were not included in the search strategy. Therefore, some relevant literature might be missed. This should be mentioned as a limitation.

Reviewer #6: The present paper presents a meta-analysis of studies measuring depression among people in Uganda. The pooled prevalence was quite high (around 35%), with even higher rates during the COVID pandemic and in the refugee’s subpopulation.

The manuscript is overall well written (except maybe for the discussion part where more wording errors were found) and is focused on a very important issue. It is great that the authors focused on Uganda because as a low-income country, it could bring results that could apply to and inform treatment in similar countries, when most of medical studies are restrained to high income countries. I think it is also very important that this kind of data is published in generalist journals such as PLOS One and not only in regional journals, in order to bring these issues to the world population.

I do have some methodological issues though that need addressing.

Major points

1. Line 205-206: “The pooled prevalence of depression was 30.2%”

This number is very different from a simple prevalence computation (27989/123859 * 100 = 22.6%, which is already huge actually). Pardon my ignorance but please provide more background for the readers to understand how you computed the pooled prevalence (I guess there is a specific weight attributed to each study, but in what way?). It would be useful to have this information in the "Data synthesis and analysis" section. Maybe the weight was attributed in part due to quality scores of the studies that were given by the authors when reviewing the articles? If this is the case it needs to be specified.

2. Line 206-207: “The funnel plot was asymmetrical”

Very asymmetrical indeed… Are you sure there was no mistake made in plotting the funnel plot? Usually, it is the low powered studies that are lower on the y axis and that are more biased towards the extremes on the x-axis. Here, all studies are at the top of the y-axis, and there are almost all highly biased toward the right. All studies but 4 exceed the triangular region, which is crazy! Overall, the data points on this plot behave in a very strange way and data should be checked again for errors. If the funnel plot is correct, how can we be confident in the pooled valence estimation with such an apparent publication bias? The heterogeneity is also very high. Can the authors identify a subgroup of studies that are maybe less biased and more coherent? If the funnel plot is correct, the publication bias must absolutely be discussed in the discussion section. Is there evidence that publication biases are higher in low-income countries for example? What could be the reason?

3. Line 273: “the estimated slope from Egger’s test was 6.12 (SE=0.598, p<0.001).”

Is it normal for this value to be the same for every subgroup analysis? Is there a mistake? The value was the same for the complete pool of studies as well. Why not report this value once if it's not a mistake ? Please double check these values or provide an explanation for why these numbers are all the same.

Minor points

1. Line 364: “The prevalence of depression ranged between 2.0% among individuals in a fishing community [83] and 68% among national humanitarian aid workers”

Are humanitarian aid workers really part of the general population? It seems to me that it's quite a stressful and special status to be part of the national humanitarian aid but maybe I have a misconception of this status. Are they confronted to stressful events? I ask this question also because the depression prevalence for this subgroup is so high compared to the fishing community.

2. Line 430 and throughout the text : “This difference may be because the present review included more studies than the former.”

I think that this assertion is uninformative. Adding more studies could either lower or increase the prevalence. They could also have no impact on the former prevalence. A more informative assertion could be that adding more studies brought the pooled prevalence closer to the "true" value, the one that could be measured in the complete population, since you are increasing the sample size (and then you are reducing the discrepancy between population and sample). It could also be due to the fact that something happened in the country in 2018 or later that changed how depression affect people in Uganda (the COVID pandemic? Another factor?).

3. Line 436 : “Future researchers should conduct reviews of studies with fewer variations especially in relation to the tools used to assess depression”.

I completely agree with this, but maybe the authors could try to do this work themself, at least in part? Maybe with a subanalysis focused on one tool that was used in the majority of the studies (for example the PHQ or the DHSCL?). It could be very interesting for readers if you could show for instance that when you take only the PHQ tool, heterogeneity is lowered drastically. This could lead to a straightforward advice of systematically use this tool for all future studies in Uganda in order to increase comparability between studies and then allow for higher quality meta-analyses? This is merely a suggestion, but it could really improve the impact of your paper, since you already collected all these data.

Reviewer #7: This is a systematic review and meta-analysis to estimate the prevalence of depression in Uganda. As the authors wrote, it would be difficult to conduct a census or cohort to estimate prevalence in low-income countries, such as Uganda. The strengths of the study include its methodology. The manuscript meets the high standard of systematic review and meta-analysis. I have a superficial comment on wording.

I thought it problematic to state “Prevalence of depression in Uganda” because they included studies with people with specific characteristics, such as HIV positive. “3.3.10. Depression among the general population” would not show the prevalence of depression in the general population given that national humanitarian aid workers had substantially high depression prevalence.

Reviewer #8: Thank you for the opportunity to review the paper entitled “Prevalence of depression in Uganda. A systematic review and meta-analysis”. Across 127 identified studies, the pooled estimate of depression was 30% in Uganda, somewhat larger than previously reported in systematic reviews (27%). Prevalence of depression was highest in people who were refugees, war victims, post-partem, living with HIV, or young in age. Prevalence of depression was also higher after the COVID-19 pandemic (48%) compared to before (29%). This paper is very well written and reemphasizes the lack of mental health needs being addressed in at-risk groups within Uganda. The methods used to conduct the systematic review and meta-analysis were appropriate and contribute to a thorough and objective evaluation of the findings across the literature. I have a couple of comments to address.

For the analyses pre and post COVID-19 pandemic, are there any third factors that could explain higher rates of depression (differences in sample characteristics between studies assessed at either period). How confident are the authors that this effect is due to the COVID-19 pandemic rather than study differences? This should be addressed to help show readers understand how best to trust such differences.

The authors noted significant evidence of publication bias and small sample study bias in many of their analyses. I commend the authors on their close examination of potential bias. I hope the authors could speak more to this in the limitations part of the discussion.

The authors bring up the high level of inconsistency across studies (Higgins I2) in the limitations section. In the limitations section, the authors should highlight the use of random effect meta-analyses which are preferred in such cases (as opposed to fixed effect meta-analyses). Also, did the authors consider meta-regressions to assess study-level predictors of variation (i.e., assessment type, population type, age of sample, gender ratio, etc.)? This could directly test many of these hypothesized contributors to variation. I think if the authors were able, a meta-regression across all the selected studies would be worthwhile.

If the authors are referring to gender (a social construct), please refer to women/men rather than female and male (female/male refers to sex).

Reviewer #9: Comments to the author

I would like to thank the editor for giving me the chance to act as an academic reviewer of this interesting work from Uganda. I am really impressed by this paper starting from its title to the way it is presented. I will have few issues as I have pointed below to be considered by the authors.

The authors should use the PRISM-P instead of the PRISMA guideline for the systematic review and meta-analysis.

The authors reported as they excluded those observational studies which were included in previous systematic review studies in the PRISMA flow diagram. This is against your justification of this study which is to determine the comprehensive pooled prevalence of depression in Uganda. I would like to the author to strongly consider this and explain how the inclusion of these studies will have a problem in their study result.

The authors reported as they have conducted sensitivity analysis since there was heterogeneity between studies. What sources of heterogeneity were identified as a result of your sensitivity analysis? Explain and better to include in the manuscript

The authors added too many figures in this paper; I would like to advise them to minimize the number of tables and figures.

Similarly, the authors used too many references. Better to limit by avoiding the outdated references used.

The discussion section is interesting but it is satisfactory as of using 173 references. Your study can best be supported by more scientific evidences in addition to your presented evidences. I strongly advice to make it stronger.

Reviewer #10: In this systematic review and meta-analysis, authors sought to determine prevalence of depression across various study populations in Uganda.

While the objectives are clearly stated and a clinically relevant question was included, it would be useful to indicate specific and focused questions regarding the subgroups of the populations being examined.

A comprehensive literature search was conducted and information sources are indicated well and search terms used seem reasonable, however it would be useful to indicate any reasonable limitations placed on the search (e.g. English language, journal, etc). Additionally, were there any attempts made at collecting unpublished data?

In terms of data abstraction, as the authors point out, the data was quite heterogenous and thus difficult to standardize—Some more details on how that translates into generalizable results would be useful. This is the biggest limitation in interpreting the results—the assessment tools in the studies may not be combinable and generalizable. However, the authors acknowledge this and have used appropriate methods to combine results and synthesize the highly heterogenous data. It would be useful to indicate if, in addition to self-report tools, there were some standard clinical interviews used to diagnose depression in the populations studied, and if any interview-based diagnosis prevalence can be obtained and compared meaningfully.

While the metanalysis is comprehensive and it provides a broader picture of prevalence of depression in various populations, it is still difficult to generalize the results to affect the clinical and treatment outcomes in a systematic manner. Future studies within these populations, as well as across wider regions in the country would be more useful in implementing treatments according to the needs (socioeconomic, cultural, refugee-status, etc).

Reviewer #11: -Thank you for the invitation to review this paper.

-Overall the article is not interesting for those who might look for data on depression at the county level within some specified population and way of analyzing it. I have a few comments as follows;

-The title needs modification, “prevalence of depression in Uganda”, in what population the review was conducted?

-Background: A systematic review and meta-analysis was carried out to determine the prevalence of depression across study populations in the country, which needs modification. Fix with your study population.

-Why do you include various populations in a single study? It is very difficult to compare unrelated studies in a single paper. Why did not you do it separately? I need sufficient reason. Some of the ideas in the paper seem comparative studies.

-I disagree with this choice of a title since it leaves out a group that is unrelated to this study and has depressive symptoms, which is a very difficult topic to declare "prevalence of depression in Uganda." Moreover, why did you decide to include only a few populations in the study? Given that the pooled prevalence of depression is (P=30%) in several included research, you cannot generalize at the national level.

6. PLOS authors have the option to publish the peer review history of their article (what does this mean?). If published, this will include your full peer review and any attached files.

Reviewer #1: **Yes: **Masahiro Banno

Reviewer #2: **Yes: **Alessio Bellato

Reviewer #3: **Yes: **martina billeci

Reviewer #4: No

Reviewer #5: No

Reviewer #6: **Yes: **Matias Baltazar

Reviewer #7: No

Reviewer #8: **Yes: **Tyler Reed Bell

Reviewer #9: No

Reviewer #10: No

Reviewer #11: No

---

## [Author Response · Author response to Decision Letter 0]

22 Aug 2022

Reference: PONE-D-22-10716, Title: Prevalence of depression in Uganda. A systematic review and meta-analysis. 

Journal: PLOS ONE

Journal requirements

Response: We have added a figshare DOI (https://doi.org/10.6084/m9.figshare.19579096.v1) for the data used in analysis. 

Response: This has been rectified based on the guidelines. 

4. We note that Figure 2 in your submission contain Map images which may be copyrighted. All PLOS content is published under the Creative Commons Attribution License (CC BY 4.0), which means that the manuscript, images, and Supporting Information files will be freely available online, and any third party is permitted to access, download, copy, distribute, and use these materials in any way, even commercially, with proper attribution. For these reasons, we cannot publish previously copyrighted maps or satellite images created using proprietary data, such as Google software (Google Maps, Street View, and Earth). For more information, see our copyright guidelines: http://journals.plos.org/plosone/s/licenses-and-copyright. We require you to either (1) present written permission from the copyright holder to publish these figures specifically under the CC BY 4.0 license, or (2) remove the figures from your submission:

Response: This map was an illustration of the study data. There are no copyright issues in relation to this illustration. 

5. Please upload a new copy of Figure 11 as the detail is not clear. Please follow the link for more information: https://blogs.plos.org/plos/2019/06/looking-good-tips-for-creating-your-plos-figures-graphics/" https://blogs.plos.org/plos/2019/06/looking-good-tips-for-creating-your-plos-figures-graphics/

Response: This has been provided as requested. 

 

Additional editors’ comments

Comment: We request that a point-by-point response letter accompanies your revised manuscript. This letter must provide a detailed response to each reviewer/editorial point raised, describing what amendments have been made to the manuscript text and where these can be found (e.g. Methods section, line 12, page 5). If you disagree with any comments raised, please provide a detailed rebuttal to help explain and justify your decision.

Response: Our detailed responses have been provided below as requested. 

Comment: The way of presenting data is needing major and detailed revision, more clarity on the analysis process can be addressed in a revision.

Response: We have made significant changes to the analysis section and the data presentation has been improved based on the multiple reviewers’ suggestions.

Comment: There are many formatting issues that may need to be addressed (many spacing issues). The Figures seem to also need to be higher quality.

Response: We have addressed the formatting issues in the revised version and the figures are now in higher quality. 

Comment: The reporting of results and discussion requires revisions to be more succinct and concise. Some reviewers were suggesting rejection or major revision.

Response: Thanks to the many reviewers’ comments, we have revised the sections based on their comments and have been more succinct and concise.

 

Comments from reviewers

Response to the Reviewer 1’s Comments 

Comment: First, in the Abstract, the authors had better adhere to PRISMA 2020 for Abstract. For example, they had better describe the following information in the abstract: inclusion criteria and exclusion criteria for the review, the date when each database was last searched, the methods used to assess the risk of bias in the included studies, the methods used to present and synthesize results, the number of participants for the prevalence of depression in refugees, war victims, individuals living with HIV, postpartum or pregnant mothers, university students, children and adolescents, caregivers of patients, a brief summary of the limitations of the evidence included in the review (e.g. study risk of bias, inconsistency and imprecision), the primary source of funding for the review, the register name and registration number.

Response: This has been updated in the revised manuscript as suggested. Abstract section, Page 3, lines 18 to 43.

Comment: Second, in Figure 1, they had better report the correct number for records. For example, the sum of records for databases is not n=513 whereas they reported the sum of records for databases is n=513. Moreover, they reported that they excluded 103 records from 216 records, then 121 records remained.

Response: The most recent version with the correct number for records has now been uploaded. Fig 1.

Comment: Third, they had better describe the results of each nine-item for JBI checklist in results. They had better describe the discussion for the results of each item for the JBI checklist in the Discussion.

Response: We now have a table listing the total scores of the JBI from the different studies. We used a cut-off (4) for studies with possible bias and poor quality. With 16% (21/127) of the studies scoring less than 9, we would prefer not to present a large table of the 127 papers showing their individual scores. However, the scores are interpreted in the revised discussion (see limitation section, page 39, lines 487 to 496). 

Response to the Reviewer 2’s Comments 

Comment: Table 1. May I ask why the articles have not been reported in alphabetical order, or what was the sorting rule used? It looks like year (oldest -> newest), but year of data collection is missing for some studies. I am not asking to change it, just to clarify, maybe in Table caption or as a note?

Response: Year of data collection was the first criterion considered for the arrangement but for those without a year of data collection, we used the year the paper was first submitted to a journal to estimate the year of data collection based on the period of data collection. Table 1 

Comment: Moreover, I think it would be good to add “(years)” to Age, as you did for the other variable and, maybe, instead of n you could report % Female?

Response: In the revised manuscript, ‘Years’ has been added as suggested, but for gender, we used absolute numbers, that is, female (male) numbers. Table 1.

Comment: At page 25, L176, you reported p<0.001, is this in relation to heterogeneity? This also applies to page 26, L206 and on all other occasions. As before, maybe it would be good to state it clearly in the text, e.g., by using a semi-colon before I2?

Response: This has been rectified as suggested in the revised manuscript. 

Comment: Please, correct “we” to “were” at L176

Response: This has been rectified in the revised manuscript as suggested. 

Comment: There is something weird in the PDF at P26, L182, could you make sure, in your doc version, that all references in relation to “Depression sub-section of 182 the Hopkins Symptom Checklist (DHSCL).” have been included?

Response: This has been rectified in the revised manuscript. Thanks for spotting this

Comment: Considering you detected publication bias, would you consider using trim and fill analysis to estimate the number of missing studies? Good that you carried out a sensitivity analysis, at L209.

Response: We have now added trim and fill analysis findings to the revised Results as suggested. Thanks for the positive feedback concerning the sensitivity analysis.

Comment: P28, could you please consider adding a statement of the results presented in Table 2, in relation to Tools used?

Response: As requested, we have now added information describing Table 2. Results section, Page 29, lines 238 and 245. 

Comment: Could you please consider improving readability in sentence “The prevalence among the elderly [140], 288 tuberculosis patients [111], post tuberculosis lung diseases [91], cancer [11], post-stroke 289 [86], physically ill [117], diabetes mellitus [119], sickle cell disease [43], and rheumatoid 290 arthritis [27] were 18%, 23.7%, 24%, 26%, 31.5%, 33.7%, 34.8%, 68.2%, and 70.8%, 291 respectively.”? Maybe adding each percentage close to the group label?

Response: This has now been rewritten as suggested in the revised manuscript.

Comment: Previous systematic review and MA (166) is only cited in the discussion, but I think it would be good to mention, in the introduction, how yours is different from previous SR/MA already published on the topic.

Response: We have now added information relating to this study in the revised Introduction. 

Comment: I have particularly appreciated the discussion on factors potentially leading to increased depression rates in Uganda, compared to other countries.

Response: Thank you for the positive feedback.

Response to the Reviewer 3’s Comments

Comment: Recently, Opio JN et al. performed a meta-analysis to determine the prevalence of mental disorders in Uganda, including major depressive disorder. However, to date, the present work is the first meta-analysis specifically addressing the prevalence of depression in Uganda, with a very large sample size and across several special populations (caregivers, individuals living with HIV or other chronic disease, etc.). The statistical analysis was correctly performed, as well as the risk of bias and the quality of the included studies were correctly assessed.

Response: Thank you for the positive feedback

Response to the Reviewer 4’s Comments 

Comment: Reviewer #4: This systematic review reports the prevalence of depression in Uganda. This review is written according to the PRISM guidelines. The authors know the limitations on the judge of depression. It is important to be interpreted.

Response: Thank you for the positive feedback.

Response to the Reviewer 5’s Comments 

Comment: 2.2 Inclusion criteria and exclusion criteria: - If there were other pre-specified of eligibility criteria, including language (e.g. English, Luganda), definition of depression (e.g. depressed mood, depressive disorder), assessment tools for depression (e.g. type, cutoff point), minimum sample size, study quality (e.g. JBI score of 4 or more), they should be mentioned in this section.

Response: This has now been added in the revised manuscript as suggested. Methods section, Page 6, lines 112 to 118.

Comment: - Having information on prevalence of depression should be mentioned as one of the inclusion criteria.

Response: This has now been added in the revised manuscript as suggested. Methods section, Page 6, line 114.

Comment: 4.0 Discussion: (Line 372-387) The finding that the pooled prevalence of depression in Uganda from the present study is higher than that from the previous study might be due to the difference in assessment tools for depression between the two studies, besides the effects of COVID-19 pandemic. Specifically, the previous review by Opio used more strict criteria for depressive disorder. Most of the included studies in Opio’s review used M.I.N.I., a structural interview, to assess depressive disorder.

Response: We included all the studies present in Opio’s review. We strongly believe (as explained in the manuscript) that the difference could be due to the additional papers added. However, we have now added the discussion of study tools in a new paragraph concerning the causes of heterogeneity. Discussion section, page 39, lines 472 to 485. 

Comment: (Lines 433-435) How “high levels of heterogeneity due to the depression assessment tools and the period of the study” affect the results/implementation of the results. This should be explained.

Responses: This has now been expanded in the revised manuscript to make it clearer for readers. Discussion section, page 39, lines 472 to 485.

Comment: (Lines 436-437) “Second, the included studies were all prone to recall biases, since all the data were self-report.” I am not sure if this statement is correct. For example, M.I.N.I. is a structured interview, not a self-report measure.

Response: Thank you for spotting this error. We have now rectified the statement in the revised manuscript. Limitation section, page 40, lines 496 to 497.

Comment: Some relevant literature in other languages (e.g. Luganda) or from other databases (e.g. EMBASE, CINAHL) might be missed. This should be mentioned as a limitation.

Response: This has been added as suggested. However, there were no academic publications written in Luganda. Limitation section, page 40, lines 501 to 504.

Comment: Some relevant terms for depression, such as mental health, psychological disorder/problem, and mood, were not included in the search strategy. Therefore, some relevant literature might be missed. This should be mentioned as a limitation.

Response: This has been added to the limitations in the revised manuscript as suggested. Limitation section, page 40, lines 504 to 505

Response to the Reviewer 6’s Comments 

Comment: Line 205-206: “The pooled prevalence of depression was 30.2%”

This number is very different from a simple prevalence computation (27989/123859 * 100 = 22.6%, which is already huge actually). Pardon my ignorance but please provide more background for the readers to understand how you computed the pooled prevalence (I guess there is a specific weight attributed to each study, but in what way?). It would be useful to have this information in the "Data synthesis and analysis" section. Maybe the weight was attributed in part due to quality scores of the studies that were given by the authors when reviewing the articles? If this is the case it needs to be specified.

Response: Thanks for the comment. A pooled prevalence is weighted based on several factors such as sample size of the individual included studies. It will rarely be as close to simple computed prevalence. We have now added a reference with information about the mathematical basis of meta-analysis pooled prevalence in the revised manuscript to guide readers. Reference 47.

Comment: Line 206-207: “The funnel plot was asymmetrical”. Very asymmetrical indeed… Are you sure there was no mistake made in plotting the funnel plot? Usually, it is the low powered studies that are lower on the y axis and that are more biased towards the extremes on the x-axis. Here, all studies are at the top of the y-axis, and there are almost all highly biased toward the right. All studies but 4 exceed the triangular region, which is crazy! Overall, the data points on this plot behave in a very strange way and data should be checked again for errors. If the funnel plot is correct, how can we be confident in the pooled valence estimation with such an apparent publication bias? The heterogeneity is also very high. Can the authors identify a subgroup of studies that are maybe less biased and more coherent? If the funnel plot is correct, the publication bias must absolutely be discussed in the discussion section. Is there evidence that publication biases are higher in low-income countries for example? What could be the reason?

Response: In addition to the sensitivity analysis we performed, we carried out further tests to elaborate on the publication bias of the included studies. The computation of all the images in this study are through a computer program and we believe there was no error in the output (they remained the same following rerunning the commands). The whole paper was based on a subgroup analysis and we also provided a prevalence rate following the sensitivity analysis, Figure 3. We have now added information in the revised Discussion to also explain the possible publication biases. 

Comment: Line 273: “the estimated slope from Egger’s test was 6.12 (SE=0.598, p<0.001).”

Is it normal for this value to be the same for every subgroup analysis? Is there a mistake? The value was the same for the complete pool of studies as well. Why not report this value once if it's not a mistake? Please double check these values or provide an explanation for why these numbers are all the same.

Response: This has been double-checked and the value has been recalculated using STATA 17. 

Comment: Line 364: “The prevalence of depression ranged between 2.0% among individuals in a fishing community [83] and 68% among national humanitarian aid workers”. Are humanitarian aid workers really part of the general population? It seems to me that it's quite a stressful and special status to be part of the national humanitarian aid but maybe I have a misconception of this status. Are they confronted to stressful events? I ask this question also because the depression prevalence for this subgroup is so high compared to the fishing community.

Response: They are indeed a special group, but we consider them as the general population because these individuals were all workers and return to their normal jobs following the activity. However, for clarity, we have now added a statement concerning our classification's limitations since different reasons may lead to these classifications. Limitation section, page 40, lines 508 to 513. 

Comment: Line 430 and throughout the text: “This difference may be because the present review included more studies than the former.” I think that this assertion is uninformative. Adding more studies could either lower or increase the prevalence. They could also have no impact on the former prevalence. A more informative assertion could be that adding more studies brought the pooled prevalence closer to the "true" value, the one that could be measured in the complete population, since you are increasing the sample size (and then you are reducing the discrepancy between population and sample). It could also be due to the fact that something happened in the country in 2018 or later that changed how depression affect people in Uganda (the COVID pandemic? Another factor?).

Response: This has now been amended in the revised manuscript as suggested. Discussion section, page 39, lines 472 to 485.

Comment: Line 436: “Future researchers should conduct reviews of studies with fewer variations especially in relation to the tools used to assess depression”. I completely agree with this, but maybe the authors could try to do this work themself, at least in part? Maybe with a subanalysis focused on one tool that was used in the majority of the studies (for example the PHQ or the DHSCL?). It could be very interesting for readers if you could show for instance that when you take only the PHQ tool, heterogeneity is lowered drastically. This could lead to a straightforward advice of systematically use this tool for all future studies in Uganda in order to increase comparability between studies and then allow for higher quality meta-analyses? This is merely a suggestion, but it could really improve the impact of your paper, since you already collected all these data.

Response: This subgroup analysis was provided in the manuscript (Table 2) but all the most commonly used instruments had high levels of heterogeneity (all with I2 above 99%). We have not gone further to recommend any particular instrument for future studies. However, we have recommended the commonly used tools to be used in future studies to increase comparability between studies and allow for higher quality meta-analyses. Limitation section, page 40, lines 492 to 494. 

Response to the Reviewer 7’s Comments 

Comment: I thought it problematic to state “Prevalence of depression in Uganda” because they included studies with people with specific characteristics, such as HIV positive. “3.3.10. Depression among the general population” would not show the prevalence of depression in the general population given that national humanitarian aid workers had substantially high depression prevalence.

Response: We used a general description to represent all the study groups and tried to show the differences by showing the prevalence rate in the different special groups. For the case of adding humanitarian workers to the general population, we have now added this as a limitation since different individuals will have different interpretations of this finding. Limitation section, page 40, lines 510 to 512.

Response to the Reviewer 8’s Comments 

Comment: For the analyses pre and post COVID-19 pandemic, are there any third factors that could explain higher rates of depression (differences in sample characteristics between studies assessed at either period). How confident are the authors that this effect is due to the COVID-19 pandemic rather than study differences? This should be addressed to help show readers understand how best to trust such differences.

Response: We have now added this perspective to the revised Discussion of the findings. Our systematic review included all the studies included in the previous review. 

Comment: The authors noted significant evidence of publication bias and small sample study bias in many of their analyses. I commend the authors on their close examination of potential bias. I hope the authors could speak more to this in the limitations part of the discussion.

Response: We have now provided a more elaborate discussion on the publication bias of the included studies in the revised limitation section. Page 40, lines 487 to 496.

Comment: The authors bring up the high level of inconsistency across studies (Higgins I2) in the limitations section. In the limitations section, the authors should highlight the use of random effect meta-analyses which are preferred in such cases (as opposed to fixed effect meta-analyses). Also, did the authors consider meta-regressions to assess study-level predictors of variation (i.e., assessment type, population type, age of sample, gender ratio, etc.)? This could directly test many of these hypothesized contributors to variation. I think if the authors were able, a meta-regression across all the selected studies would be worthwhile.

Response: Our meta-analysis was based on the preferred random effect model (limitation section, page 40, line 489). Also, a meta-regression has now been added to the revised manuscript as suggested. 

Comment: If the authors are referring to gender (a social construct), please refer to women/men rather than female and male (female/male refers to sex).

Response: Different studies were not consistent concerning gender or sex. We did not generalize the terms used but collected the data presented, capturing males and females as shown in Table 1. 

Response to the Reviewer 9’s Comments 

Comment: The authors should use the PRISM-P instead of the PRISMA guideline for the systematic review and meta-analysis.

Response: Thank you for your comment but our study was not a protocol. 

Comment: The authors reported as they excluded those observational studies which were included in previous systematic review studies in the PRISMA flow diagram. This is against your justification of this study which is to determine the comprehensive pooled prevalence of depression in Uganda. I would like to the author to strongly consider this and explain how the inclusion of these studies will have a problem in their study result.

Response: These studies were included in our study. This is clearly shown in the PRISMA flow chart. Fig 1.

Comment: The authors reported as they have conducted sensitivity analysis since there was heterogeneity between studies. What sources of heterogeneity were identified as a result of your sensitivity analysis? Explain and better to include in the manuscript

Response: In the revised manuscript, we added further analysis to assess for publication bias and elaborated on the causes of heterogeneity. 

Comment: The authors added too many figures in this paper; I would like to advise them to minimize the number of tables and figures.

Response: Based on the comments from other reviewers, we have retained most of the figures to clearly show the readers the results and to avoid any misinterpretation. 

Comment: Similarly, the authors used too many references. Better to limit by avoiding the outdated references used.

Response: We have reviewed the references and made sure we have updated references and some were eliminated. The many references are mainly due to the many studies included in the review. 

Comment: The discussion section is interesting but it is satisfactory as of using 173 references. Your study can best be supported by more scientific evidences in addition to your presented evidences. I strongly advice to make it stronger.

Response: We have added more evidence-based information to the revised Discussion. 

Response to the Reviewer 10’s Comments 

Comment: In this systematic review and meta-analysis, authors sought to determine prevalence of depression across various study populations in Uganda.

Response: Thank you for your kind observations and suggestions to us. These helped us to strengthen the manuscript. 

Comment: While the objectives are clearly stated and a clinically relevant question was included, it would be useful to indicate specific and focused questions regarding the subgroups of the populations being examined.

Response: We have now added a specific question about the subgroups. 

Comment: A comprehensive literature search was conducted and information sources are indicated well and search terms used seem reasonable, however it would be useful to indicate any reasonable limitations placed on the search (e.g. English language, journal, etc). Additionally, were there any attempts made at collecting unpublished data?

Response: No. We have now added this to the limitations in the revised manuscript. 

Comment: In terms of data abstraction, as the authors point out, the data was quite heterogenous and thus difficult to standardize—Some more details on how that translates into generalizable results would be useful. This is the biggest limitation in interpreting the results—the assessment tools in the studies may not be combinable and generalizable. However, the authors acknowledge this and have used appropriate methods to combine results and synthesize the highly heterogenous data. It would be useful to indicate if, in addition to self-report tools, there were some standard clinical interviews used to diagnose depression in the populations studied, and if any interview-based diagnosis prevalence can be obtained and compared meaningfully.

Response: A subgroup classification has now been added based on the diagnostic status of the study tool. See Table 2. 

Comment: While the metanalysis is comprehensive and it provides a broader picture of prevalence of depression in various populations, it is still difficult to generalize the results to affect the clinical and treatment outcomes in a systematic manner. Future studies within these populations, as well as across wider regions in the country would be more useful in implementing treatments according to the needs (socioeconomic, cultural, refugee-status, etc).

Response: We have now added this as a recommendation in the revised manuscript. 

Response to the Reviewer 11’s Comments 

Comment: The title needs modification, “prevalence of depression in Uganda”, in what population the review was conducted?

Response: None of the other 10 reviewers had any problem with the title and we also believe we have followed the examples of many other similar studies in the literature. Therefore, we have retained our original title. 

Comment: Background: A systematic review and meta-analysis was carried out to determine the prevalence of depression across study populations in the country, which needs modification. Fix with your study population.

Response: We have specified that we included different study populations in the country as shown in the Results section of the revised Abstract (Abstract section, page 2, lines 21 and 22). 

Comment: Why do you include various populations in a single study? It is very difficult to compare unrelated studies in a single paper. Why did not you do it separately? I need sufficient reason. Some of the ideas in the paper seem comparative studies.

Response: The pooled prevalence of depression in Uganda was the focus of our study. For this purpose, all the studies (whether they are unrelated or not in terms of population) have to be included. We have followed standard protocol for estimating pooled prevalence. Despite the different populations, the estimated prevalence rate from this study is helpful for understanding the overall situation of depression in the country. Synthesizing data from such diverse studies is standard practice in these types of meta-analysis

Comment: I disagree with this choice of a title since it leaves out a group that is unrelated to this study and has depressive symptoms, which is a very difficult topic to declare "prevalence of depression in Uganda." Moreover, why did you decide to include only a few populations in the study? Given that the pooled prevalence of depression is (P=30%) in several included research, you cannot generalize at the national level.

Response: We refer you to our previous responses. The title captures what we have done and we have followed the usual procedure for estimating pooled prevalence rates.

---

## [Decision Letter · Decision Letter 1]

10 Oct 2022

Prevalence of depression in Uganda: A systematic review and meta-analysis

PONE-D-22-10716R1

Dear Dr. Kaggwa,

We’re pleased to inform you that your manuscript has been judged scientifically suitable for publication and will be formally accepted for publication once it meets all outstanding technical requirements.

Kind regards,

Muhammed Elhadi, MBBCh

Academic Editor

PLOS ONE

Additional Editor Comments (optional):

Reviewers' comments:

Reviewer's Responses to Questions

**Comments to the Author**

1. If the authors have adequately addressed your comments raised in a previous round of review and you feel that this manuscript is now acceptable for publication, you may indicate that here to bypass the “Comments to the Author” section, enter your conflict of interest statement in the “Confidential to Editor” section, and submit your "Accept" recommendation.

Reviewer #1: All comments have been addressed

Reviewer #2: All comments have been addressed

Reviewer #5: All comments have been addressed

Reviewer #6: All comments have been addressed

Reviewer #7: All comments have been addressed

Reviewer #8: All comments have been addressed

Reviewer #9: All comments have been addressed

Reviewer #11: All comments have been addressed

2. Is the manuscript technically sound, and do the data support the conclusions?

Reviewer #1: Yes

Reviewer #2: Yes

Reviewer #5: Yes

Reviewer #6: Yes

Reviewer #7: Yes

Reviewer #8: Yes

Reviewer #9: Yes

Reviewer #11: Yes

3. Has the statistical analysis been performed appropriately and rigorously? 

Reviewer #1: Yes

Reviewer #2: Yes

Reviewer #5: Yes

Reviewer #6: Yes

Reviewer #7: No

Reviewer #8: Yes

Reviewer #9: Yes

Reviewer #11: Yes

4. Have the authors made all data underlying the findings in their manuscript fully available?

Reviewer #1: Yes

Reviewer #2: Yes

Reviewer #5: Yes

Reviewer #6: Yes

Reviewer #7: Yes

Reviewer #8: Yes

Reviewer #9: Yes

Reviewer #11: Yes

5. Is the manuscript presented in an intelligible fashion and written in standard English?

Reviewer #1: Yes

Reviewer #2: Yes

Reviewer #5: Yes

Reviewer #6: Yes

Reviewer #7: Yes

Reviewer #8: Yes

Reviewer #9: Yes

Reviewer #11: No

6. Review Comments to the Author

Reviewer #1: The authors have revised the manuscript thoroughly. I have no additional comments about the manuscript.

Reviewer #2: Thanks for addressing all comments.

Paper is now suitable for publication, in my opinion.

Interesting and methodologically correct.

Reviewer #5: The authors have addressed all my comments. I am satisfied with the revisions made on the manuscript.

Reviewer #6: The reviewers have responded satisfactorily to all my comments. I am still puzzled by the very asymetrical funnel plot but the authors assure that they rerun the analysis and that all is correct so I guess this is the reflection of a strong publication bias. Fortunately, the authors address this issue in the discussion section. I thank the authors for all their hard work.

Reviewer #7: The authors did their best to respond to the reviewers' comments. I have no concerns on research ethics.

Reviewer #8: Thank you for addressing my comments. I believe the paper is substantially improved. Results highlight an important public health target, i.e., depressive symptoms, in Uganda.

Reviewer #9: I would like to say thank you to the author for addressing my concerns. All is about the improvement of the paper for its better quality as it is a scientific paper.

Reviewer #11: The manuscript has a significant improvement. It has a good scientific contribution. But, some grammatical issues should be resolved.

7. PLOS authors have the option to publish the peer review history of their article (what does this mean?). If published, this will include your full peer review and any attached files.

Reviewer #1: **Yes: **Masahiro Banno

Reviewer #2: No

Reviewer #5: No

Reviewer #6: **Yes: **Matias Baltazar

Reviewer #7: No

Reviewer #8: **Yes: **Tyler Reed Bell

Reviewer #9: No

Reviewer #11: No

---

## [Editor Report · Acceptance letter]

12 Oct 2022

PONE-D-22-10716R1 

Prevalence of depression in Uganda: A systematic review and meta-analysis 

Dear Dr. Kaggwa:

I'm pleased to inform you that your manuscript has been deemed suitable for publication in PLOS ONE. Congratulations! Your manuscript is now with our production department. 

Kind regards, 

on behalf of

Dr. Muhammed Elhadi 

Academic Editor

PLOS ONE